# Dynamic anticrack propagation in snow

J. Gaume [1,2], T. Gast[3,4], J. Teran[3,4], A. van Herwijnen [2] & C. Jiang[4,5]

Continuum numerical modeling of dynamic crack propagation has been a great challenge over the past decade. This is particularly the case for anticracks in porous materials, as reported in sedimentary rocks, deep earthquakes, landslides, and snow avalanches, as material inter-penetration further complicates the problem. Here, on the basis of a new elastoplasticity model for porous cohesive materials and a large strain hybrid Eulerian–Lagrangian numerical method, we accurately reproduced the onset and propagation dynamics of anticracks observed in snow fracture experiments. The key ingredient consists of a modified strain-softening plastic flow rule that captures the complexity of porous materials under mixed-mode loading accounting for the interplay between cohesion loss and volumetric collapse. Our unified model represents a significant step forward as it simulates solid-fluid phase transitions in geomaterials which is of paramount importance to mitigate and forecast gravitational hazards.

[1] School of Architecture, Civil and Environmental Engineering, Swiss Federal Institute of Technology, 1015 Lausanne, Switzerland. [2] WSL Institute for Snow and Avalanche Research SLF, Flüelastrasse 11, Davos Dorf, Switzerland. [3] Department of Mathematics, University of California, Los Angeles, CA 90095, USA. [4] Jixie Effects, Los Angeles, CA 90095, USA. [5] Computer and Information Science Department, University of Pennsylvania, Philadelphia, PA 19104, USA. Correspondence and requests for materials should be addressed to J.G. (email: johan.gaume@gmail.com)

Cohesive porous materials under compression often evidence volumetric collapse leading to localization of compaction or compacting shear bands[1,2]. This peculiar fracture process is generally referred to as anticrack and is reported in the compression of porous sandstone and sedimentary rocks[3,4], superheated ice[5], submarine landslides[6], deep earthquakes[7,8] as well as in brittle foams[9]. Anticrack propagation is also believed to be at the origin of dangerous dry snow slab avalanches[10] that are responsible for most avalanche accidents. Slab avalanches originate due to the mixed-mode failure of a porous weak snow layer buried below a dense and cohesive snow slab[11]. Once the initial failure reaches a critical size, the fracture propagates along the slope possibly leading to the detachment and sliding of the overlying slab if the slope-parallel gravitational force overcomes friction[12]. While such avalanches were for a long time believed to initiate due to mode II shear fracture[13], recent experiments reporting fracture propagation on flat terrain as well as observations of remote avalanche triggering[14,15] challenged classical theories. This contradiction highlighted the crucial role of the cohesion loss and volumetric collapse of the porous structure of the weak layer which is generally accompanied by a so-called "whumpf" sound, indicator of snowpack instability.

Heierli et al.[10] proposed a mixed-mode (I/II) anticrack theory to characterize the conditions for the onset of crack propagation in snow slab avalanches. More recently, Gaume et al.[16,17] proposed the shear-collapse model (SCM) which improved the latter by accounting for dynamics and a more realistic mechanical behavior of the porous weak layer using the discrete element method. However, the static and discrete nature of these two models prevents upscaling to the scale of typical avalanche slopes for which a dynamic continuum approach is necessary.

In classical continuum methods for fracture[18,19] as well as in standard materials, the concept of anticrack is physically impossible due to mesh or material inter-penetration induced by volume loss. Hence, these methods are suitable for tensile and shear fractures only. In addition, existing models based on critical state soil mechanics (CSM) fail in reproducing the post-peak strain-softening behavior of porous cohesive materials since only hardening is allowed in compression. To account for cohesion loss and volume reduction in a finite element snow model, Mahajan et al.[20] artificially removed mesh elements after failure and allowed for frictional contacts of closing crack faces. Yet, so far, no standalone continuum constitutive model exists to simulate dynamic anticrack propagation in porous cohesive materials.

Here, we propose to address this crucial gap through a new elastoplastic constitutive model for porous cohesive materials that accounts for cohesion softening and volume reduction. Simulations are performed using the Material Point Method (MPM)[21], a hybrid Eulerian–Lagrangian method suitable to deal with large strains. This method is highly relevant for processes involving fractures and collisions[22–24]. Our new model accurately reproduces the onset and dynamics of propagating anticracks monitored in snow fracture experiments using high-speed photography and particle tracking. Finally, we show that our unified model simulates both the release and flow of slab avalanches at the slope scale.

## Results

**Large-strain elastoplastic model.** To model the observed process of anticrack propagation in snow, we developed a large-strain elastoplastic model. Material deformation is characterized by the strain measure. Assuming there is a deformation map $\phi(\mathbf{X}, t)$ that maps undeformed coordinate $\mathbf{X}$ to a deformed coordinate $\mathbf{x}$, the deformation gradient $\mathbf{F}$ is defined as $\partial\phi/\partial\mathbf{X}$. Our physical model assumes finite strain elastoplasticity, where $\mathbf{F}$ is decomposed into elastic ($\mathbf{F}^E$) and plastic ($\mathbf{F}^P$) parts as $\mathbf{F} = \mathbf{F}^E\mathbf{F}^P$ (multiplicative elastoplasticity). The elastic deformation gradient is computed using the isotropic Hooke's law of elasticity (see Methods section for more details).

For plasticity, the yield function $y(\boldsymbol{\tau}) \leq 0$ defines admissible stress states in an elastoplastic continuum. We model snow based on the critical state plasticity theory for soil mechanics[25,26]. For any stress $\boldsymbol{\tau}$, there exist a mean effective stress (or pressure) $p$ and a deviatoric stress $\mathbf{s}$. They are given by

$$p = -\frac{1}{d}\mathrm{tr}(\boldsymbol{\tau}), \tag{1}$$

$$\mathbf{s} = \boldsymbol{\tau} + p\mathbf{I}, \tag{2}$$

respectively, where $d = 2$ or $3$ is the problem dimension, $\mathbf{I}$ is the identity matrix and compression corresponds to $p > 0$. According to the Von Mises theory[27], we can derive the Mises equivalent stress $q$, given by $q = (3/2\ \mathbf{s} : \mathbf{s})^{1/2}$ (so that $q = |\tau_1 - \tau_2|$ for 2D and $q = \sqrt{\frac{1}{2}\left((\tau_1 - \tau_2)^2 + (\tau_2 - \tau_3)^2 + (\tau_3 - \tau_1)^2\right)}$ for 3D, in principal stress space).

Recent experiments[11] and simulations based on X-ray microtomography[28–31] highlighted the mixed-mode nature of snow failure including tensile, shear and compression failure modes. Given these past studies, it appears that an ellipsoid yield function is appropriate to reproduce this mixed-mode character. Hence we chose to start from the modified cam clay (MCC) yield surface[32] which has been widely used in the area of soil mechanics. Note that the analogy between snow and clay was already made by McClung[13] who extended the clay model of Palmer and Rice[33] to model shear fractures induced by strain softening. However, the MCC model is originally cohesionless and does not exhibit any stress under extension, similar to dry sand. Hence, cohesion was added to the yield function by shifting the MCC model along the $p$-axis. We thus propose a new cohesive cam clay (CCC) model similar to that of Meschke et al.[34] with the following yield surface:

$$y(p, q) = q^2(1 + 2\beta) + M^2(p + \beta p_0)(p - p_0), \tag{3}$$

where $p_0$ represents the consolidation pressure and $M$ is the slope of the cohesionless critical state line (CSL) that controls the amount of friction, $\beta$ represents the ratio between tensile and compressive strength and controls the amount of cohesion ($\beta \geq 0$). This yield surface is represented in Fig. 1a. Both MCC and our model are ellipsoids and are symmetric around the hydrostatic axis.

For the dense snow slab, the hardening and softening is modeled by expanding and shrinking the yield surface which is performed by varying $p_0$. We assume the hardening and softening only depend on the volumetric plastic deformation $\epsilon_V^P = \log(\det(\mathbf{F}^P))$. We follow the derivation from Ortiz and Pandolfi[35] and use the following hardening law:

$$p_0 = K\sinh\left(\xi\max\left(-\epsilon_V^P, 0\right)\right), \tag{4}$$

where $\xi$ is the hardening factor and $K$ is the material bulk modulus. When the plastic deformation is compressive $\left(\dot{\epsilon}_V^P < 0\right)$, $p_0$ will increase, causing the yield surface to grow in size. Snow will consequently receive more elastic responses resisting compression. When the plastic volume is increased $\left(\dot{\epsilon}_V^P > 0\right)$, the yield surface shrinks which allows the snow to fracture in tension. This hardening law is represented in Fig. 1b (in black).

Classical hardening/softening laws such as the one described above for the dense snow slab (Eq. 4) fail in reproducing the collapse of porous cohesive materials under compression. This is

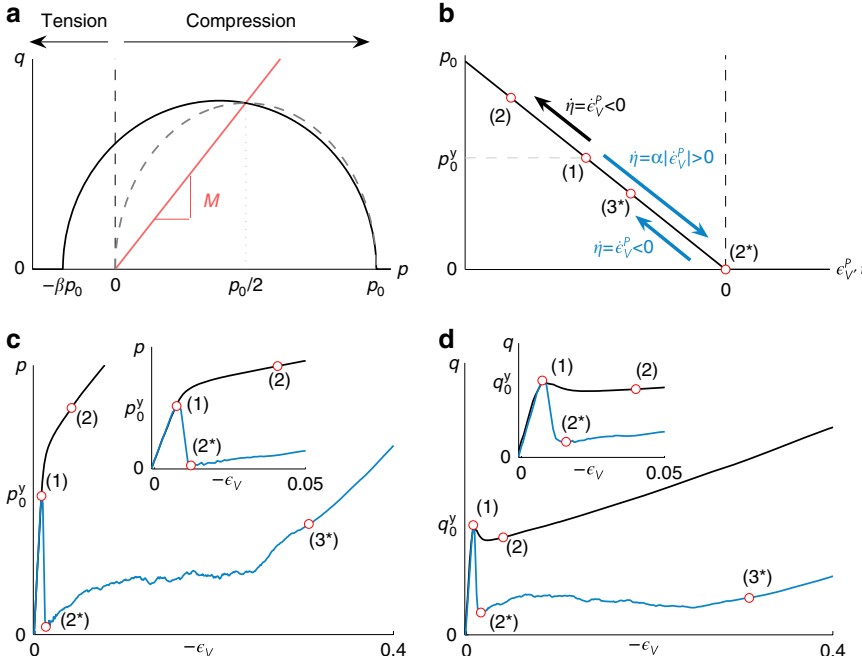

**Fig. 1** Overview of the elastoplastic model. **a** Cohesive (black line) and cohesionless (dashed gray line) cam clay yield surface in the $p$–$q$ space. The red line corresponds to the Critical State Line. **b** Illustration of the hardening models $p_0(\epsilon_V^P)$ (for the slab) and $p_0(\eta)$ (for the weak layer): the black arrow shows the classical hardening law used for the snow slab in which $p_0$ increases in compression $(\dot\epsilon_V^P<0)$; the blue arrows represent the new softening model for the weak layer for which $p_0$ decreases under compression $(\dot\eta=\alpha|\dot\epsilon_V^P|>0)$ until $\epsilon_V^P=0$ after which the classical hardening law is used with $\beta=0$. **c** Typical $p$–$\epsilon_V$ curve obtained for the unconfined compression of the weak layer in experiment number 2 (see Methods section for model parameters) for the classical hardening law (in black) and the new softening one (in blue). **d** Same as **c** but for the $q$–$\epsilon_V$ curve. In **c** and **d**, $p$ and $q$ in the weak layer (blue curves) do not perfectly reach zero after softening due to a loss of homogeneity (failure localization)

shown in Fig. 1c, d (black lines) in which $p$ significantly increases after reaching the yield surface and $q$ slightly decreases before increasing. Hence, for the porous weak layer, we propose a modified softening law that describes cohesion and volume loss under compressive stresses. This new softening law involves looking at the volumetric plastic strain rate $\dot\epsilon_V^P$. We introduce the anticrack plastic strain $\eta$ which is related to $\epsilon_V^P$ as follows:

$$\dot\eta = \begin{pmatrix} \alpha|\dot\epsilon_V^P|, & \text{if } t \leq t_c \\ \dot\epsilon_V^P, & \text{if } t>t_c \end{pmatrix} \quad (5)$$

where $\alpha$ is a softening factor which controls the energy dissipated during fracture and $t_c$ is the time corresponding to complete softening, i.e., $\epsilon_V^P=0$ and $p_0=0$ (state (2*) in Fig. 1). Our new softening law for the weak layer is obtained by replacing $\epsilon_V^P$ by $\eta$ in Eq. (4) (the discretization is shown in the Methods section). Hence, when stresses in the weak layer reach the yield surface, the introduction of the norm of $\dot\epsilon_V^P$ in Eq. (5) will lead to softening (through a decrease in $p_0$) even under compression for which $\dot\epsilon_V^P<0$. The yield surface thus shrinks until it corresponds to a point at the origin of the $p$–$q$ space. In addition, cohesion is removed by setting $\beta=0$ when $\epsilon_V^P=0$ which ensures continuity. After reaching this point, the yield surface is free to expand according to the classical hardening law (Eq. 4), leading to volume reduction (collapse) due to the weight of the slab (blue arrows in Fig. 1b) and then a purely frictional/compaction behavior. Our softening rule reproduces bond breaking in the weak layer and subsequent grain rearrangement leading to volumetric collapse due to the compressive weight of the slab[36]. In contrast to classical hardening laws, our new formulation induces a strain-softening behavior even under macroscopic uniaxial compression, as shown in Fig. 1c, d. The observed

mechanical behavior is very similar to that reported in discrete element simulations of porous cohesive granular materials[37] and follows the following sequence of mechanical regimes: elastic regime, failure, drop in pressure and shear stress (strain softening), plastic consolidation corresponding to the volumetric collapse and, finally, dense packing regime corresponding to the jamming transition. This typical post-peak behavior was also observed in laboratory experiments of snow failure[38,39] as well as during the propagation of compaction bands in confined compression of snow[2]. Physically, this behavior is related to the fact that even under a macroscopic compressive loading mode, the solid matrix of porous cohesive materials is mostly under tension (bending) and shear[37]. The behavior of the weak layer during a shear test simulation is shown in Supplementary Note 2.

To remove mesh dependency induced by softening, we follow the suggestion of Mahajan et al.[20] and Sulsky and Peterson[40] to regularize the jump in displacement. It is performed by dissipating the same amount of energy for different mesh resolutions by setting the softening factor $\alpha$ in Eq. (5) proportional to the mesh size $dx$. The influence of the mesh resolution on the volumetric plastic deformation during weak layer collapse is shown in Supplementary Note 3. For more detail about the Material Point Method and the implementation of the plastic model (plastic flow rule and return mapping), please see the Methods section.

Let us summarize here the different model parameters and their physical meaning: $p_0$ is the consolidation pressure and represents the compressive strength of the material, $\beta$ is the ratio between tensile and compressive strength and represents cohesion, $M$ is the slope of the Critical State Line (CSL) and characterizes the friction of the material, $K$ is the bulk elastic modulus, $\xi$ is the hardening coefficient and characterizes the brittleness of the material (a large $\xi$ makes snow more brittle)

and $\alpha$ is the softening factor which controls the fracture energy of the weak layer.

**Field experiments**. We report anticrack propagation in Propagation Saw Test (PST) experiments[14]. A PST consists in creating an artificial crack of increasing size by cutting within the weak layer with a saw until crack propagation. Depending on snowpack properties, the crack can either propagate until the end of the column ("END" case) or induce a fracture in the slab thus arresting the propagation ("SF" case). Black markers are inserted in the snowpack in order to derive the displacements using particle tracking velocimetry (PTV) and a high speed video camera. Two experiments were performed on flat terrain ($\psi = 0°$) and one on a typical avalanche slope ($\psi = 37°$). The density of the slab ranged from 159 to 279 kg m$^{-3}$, slab thickness ranged from 26 to 75 cm and weak layer thickness ranged from 1 to 15 cm. For more detail about the experimental set-up, snowpack properties and data analysis, please refer to the Methods section.

Figure 2 shows the vertical displacements $u_y$ of the markers during the experiments as well as the displacement field at different key instants of the experiments. For each experiment, the crack in the weak layer induces slab bending leading to relatively small displacements. After reaching the critical crack length, the vertical displacement increased significantly due to dynamic anticrack propagation inducing the progressive collapse of the weak layer.

The first experiment on the flat (experiment number 1) highlights the potential of remote avalanche triggering from low-angle terrain. All markers show significant vertical displacements (between 2.5 and 8 mm) and the fracture in the weak layer propagated until the end of the beam (END). The second experiment (experiment number 2) made on a typical avalanche slope is also a case of full propagation (END) with significant collapse of the weak layer (up to 1 cm). In this case, crack propagation is followed by the sliding of the slab since slope angle is larger than the friction angle of snow (~30°, van Herwijnen et al.[41]). Sliding induces the progressive erosion of the weak layer and thus further vertical displacement. Full propagation in the weak layer is typical for deep and dense slab layers[16]. The third experiment (experiment number 3) on the flat is a case of partial propagation in the weak layer with slab fracture (SF). In this case, markers located on the right side of the fracture did not move. This is a typical outcome for low density and shallow slab layers[15,16,42]. Movies of the three experiments, including displacement fields are provided in the Supplement (Supplementary Movies 1–3).

**PST simulations**. The hybrid Eulerian–Lagrangian Material Point Method was used to solve the set of partial differential equations of the system, given the same characteristics and boundary conditions as in the experimental PSTs. We discuss in detail the choice of snowpack mechanical properties in the Methods section.

As shown in Fig. 2 and in the Supplementary Movies, our model accurately reproduces all the features observed in the experiments. More specifically, anticrack propagation on flat terrain i.e. without external driving shear forces, is very well captured (Fig. 2a and Supplementary Movie 1). A measured critical crack length $a_c = 39$ cm was well reproduced by the simulation. The collapse wave speed $c$ was computed from the time-delay between the onset of movement between markers[15]. It was around 35 m s$^{-1}$ in both the experiment and the simulation. This speed is significantly lower than the speed of elastic waves $c^e = \sqrt{E/\rho}$ in the slab which is around 200 m s$^{-1}$.

Once the crack has propagated through the full system length, the system is at rest. Figure 2b and Supplementary Movie 2 show the results on a typical avalanche slope of 37°. Anticrack propagation features are very similar as on the flat but the propagation speed is lower ($c = 23$ m s$^{-1}$) due to a lower slab elastic modulus and density and a larger weak layer strength[16] than in experiment number 1. The bending phase, critical crack length ($a_c = 32$ cm), anticrack propagation as well as the frictional sliding of the slab are very well reproduced by our model. Crack branching resulting from the interplay between weak layer and slab fracture is also well reproduced, as shown on Fig. 2c and Supplementary Movie 3. In this case, anticrack propagation in the weak layer was arrested 10 cm after reaching a critical length ($a_c = 26.5$ cm) as the tensile stress in the slab induced by slab deformation exceeded the tensile strength due to its thin and weak character (low density slab). In contrast, in the two previous experiments, the tensile stress in the slab remained lower than the strength thus leading to full propagation. Nevertheless, we note that the bending deformation pre-propagation was underestimated by our model for experiment number 3. This suggests that inelastic (probably rate-dependent) deformation is induced by the very loose character of the slab in this experiment ($\rho_3 = 159$ kg m$^{-3}$). In addition, we observe small oscillations in our displacements because our simulations are performed without damping.

Note that for all simulations, the anticrack velocity was found almost equal to the collapse speed obtained from the vertical displacement of the slab. However, we observed that the anticrack tip is always located slightly ahead of the collapse wave front.

**Slope-scale simulations**. Two- and three-dimensional slope-scale simulations of remote avalanche triggering were performed (Supplementary Movies 4–7). In both 2D and 3D slope simulations, the average crack propagation speed was around 60 m s$^{-1}$ and the crown fracture was almost perpendicular to the bed surface as reported by Perla[43] and McClung and Schweizer[44]. Furthermore, the slab fracture at the crown of the avalanche (upslope section of the fracture line) started branching from the bottom of the slab at the interface with the weak layer (Supplementary Movie 5), in contrast to the PST simulation and experiment number 3 in which it started branching from the top. In 3D, the simulated release zone (Fig. 3, Supplementary Movie 7) has commonly observed characteristics[43]: an arc crown line as well as jagged flanks (side sections of the fracture line) and staunchwall (bottom section of the fracture line). Crown fracture occurs in tension while flank and staunchwall fractures occur in shear. Finally, the cross-slope propagation was approximately twice slower than up-slope propagation.

**Discussion**

Our new model overcomes one of the major shortcomings of Critical State Soil Mechanics, namely that it performs very poorly with materials that exhibit significant strain-softening and void ratio changes with strain[45]. It reproduced the observed failure behavior of weak snow layers, one of the most porous geomaterials (volume fraction <20%). Yet, our model can be applied to different porous media exhibiting similar behaviors, i.e., strain softening and volume reduction under compressive stresses. For instance, it has great perspectives of applications in different fields reporting anticrack fracture modes, such as in the compression of porous sandstone and sedimentary rocks, landslides as well as deep earthquakes.

Our model reproduced dynamic propagation of anticracks in porous layers of snow as well as crack branching in the case of loose and soft overlying snow slabs. More generally, our unified

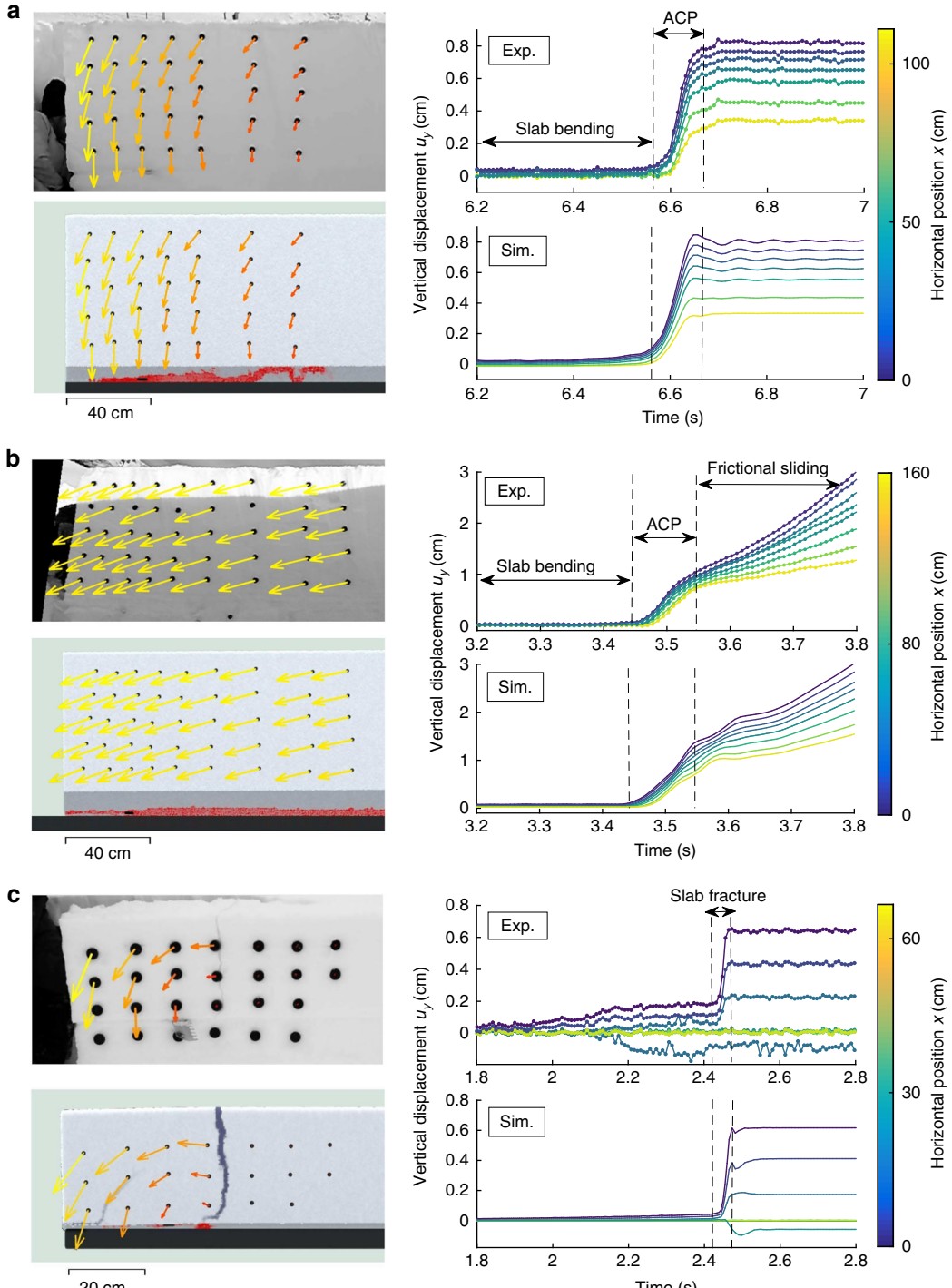

**Fig. 2** Comparison between experimental and simulated results. Experimental (top) and numerical (bottom) results for experiment number 1 (**a**), experiment number 2 (**b**), and experiment number 3 (**c**). The displacement field is shown on the left for different key instants in each case: during anticrack propagation ($t = 6.65$ s) in **a**; during frictional sliding ($t = 3.7$ s) in **b**, and after slab fracture ($t = 2.5$ s) in **c**. On the right, the time evolution of the average vertical displacement of vertical rows of markers is shown. The color represent the average horizontal position of each vertical row of markers. ACP anticrack propagation. The red color in the weak layer represents plasticity

model is relevant to simulate solid–fluid phase transitions in geomaterials. Indeed, we can simulate not only the initiation but also the flow of gravitational mass movements using a single and adequate framework as shown in the Supplementary Movie 4. This simulation corresponds to one of the most complex phenomenon in snow science, namely the remote triggering of a slab avalanche by a skier (simulated as a snowman)[46]. The skier

initiates a crack in the weak layer that propagates along the slope as a mixed-mode anticrack. The progressive loss of support of the slab leads to the release and flow of the avalanche which eventually buries the skier. Note that close to the skier, we observe local slab fractures similar to the "shooting cracks" which are reported when the avalanche danger level is considerable or higher[47]. This slope simulation reproduced so-called

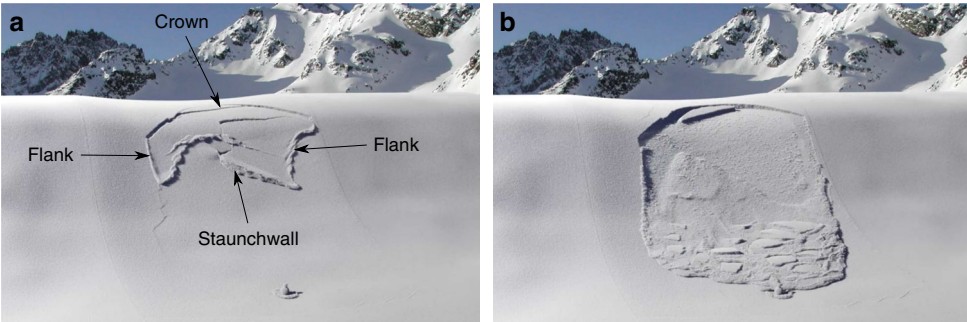

**Fig. 3** 3D slope-scale simulation of remote avalanche triggering (Supplementary Movie 7). **a** Release zone showing an arc crown line as well as jagged flanks and staunchwall. **b** Flow of the avalanche

"en-echelon" fractures during anticrack propagation which are often observed in the field[48].

Finally, there is a debate about crack branching in the slab on whether it should start from the bottom or from the top due to slab bending. We systematically observed slab fractures opening from the top in PST simulations and field experiments. However, for slope simulations (Supplementary Movies 4–7) the crown fracture always started branching at the bottom of the slab at the interface with the weak layer in agreement with near-infrared crown fracture measurements at the origin of this debate[49,50]. Hence, our model reconciles contradictory observations of slab fractures from small scale field tests (top to bottom) and from real avalanches (bottom to top). We suggest that the main reason for this discrepancy is related to the slope angle gradient at the crown and the frictional sliding of the slab. In the Supplementary Movie 5 (crown fracture in a 2D slope simulation), it appears that the crown fracture is a secondary process occurring after the crack in the weak layer has passed. Hence, the tensile stress induced by slab bending was not sufficient to induce a tensile fracture, very likely due to the large propagation speed which reduces bending as suggested by Gaume et al.[16]. However, after crack propagation and collapse, the weak layer has a frictional shear behavior leading to a pure tension stress state (no bending) in the slab[51] that has started to slide on the weak layer only where the slope is steep enough. This induces very large tensile stresses in the slab which are maximum at the interface with the weak layer where the fracture initiates. In contrast, in PST experiments and simulations (Fig. 2c), the bending of the slab induced by the crack in the weak layer created with the saw is sufficient to lead to slab fracture and the arrest of crack propagation in the weak layer. In that case, stresses are larger at the top of the slab where the fracture initiates[16].

In the future, the parameters of our model should be systematically derived from in situ measurements and related to snow type and density. The main difficulty lies in the thin and fragile nature of weak layers which prevents efficient mechanical testing such as triaxial tests to evaluate relevant model parameters. Hence, a calibration based on PST results using a larger dataset similar to what was done in van Herwijnen et al.[52] or an evaluation based on X-ray computed tomography[28] will be required. This would allow to develop a predictive model to mitigate and forecast real-scale gravitational hazards by using digital elevation models of real slopes obtained from laser scanning or photogrammetry[53] as input.

## Methods

**Experimental set-up.** Data were collected in Winter 2015–2016 in Davos, Switzerland. At each site, we collected a manual snow profile and conducted the PST according to the procedure outlined in Greene et al.[54] (Fig. 4). The PST was filmed using a high speed camera on a tripod in order to evaluate the motion of

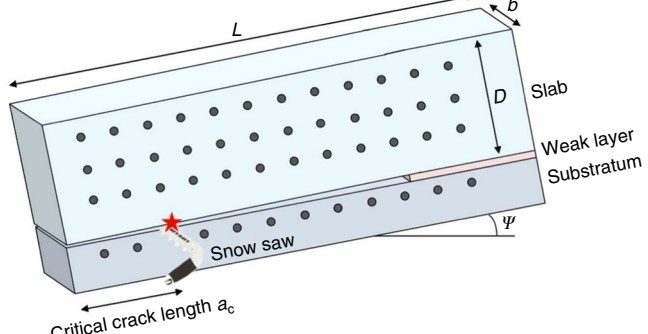

**Fig. 4** Illustration of the experimental set-up of the Propagation Saw Test (PST). After reaching the critical crack length (red star), the crack propagates along the weak layer. Black markers are inserted in the slab and the substratum to track their positions using Particle Tracking Velocimetry (PTV)

### Table 1 Parameters obtained in the experiments

| Parameter | Exp_01 | Exp_02 | Exp_03 |
|---|---|---|---|
| Slope angle $\psi$ (°) | 0 | 37 | 0 |
| Mean slab density $\rho$ (kg m$^{-3}$) | 279 | 255 | 159 |
| Slab thickness $D$ (cm) | 70 | 75 | 26 |
| Weak layer thickness $D_{wl}$ (cm) | 7.5 | 15 | 1 |
| PST outcome | END | END | SF |
| Critical crack length $a_c$ (cm) | 39 | 32 | 26.5 |
| Position of slab fracture $x_{SF}$ (cm) | — | — | 36 |
| Frames per second | 120 | 120 | 120 |

END: full propagation in the weak layer, SF: partial propagation with slab fracture

black plastic markers inserted into the pit wall using particle tracking velocimetry (PTV)[15]. This allowed us to compute the displacement of the snow slab above the weak layer with a mean accuracy of 0.1 mm. The crack propagation speed $c$ was then evaluated by computing the ratio between the horizontal distance and the time delay between the onset of vertical movement of subsequent markers as described in van Herwijnen and Jamieson[36]. Data for each test are presented in Table 1.

**Model parameters.** The parameters measured in the experiments were directly used as input of the model (geometry and density). For the slab, the Young's modulus $E$ and tensile strength $\beta p_0$ were derived from density based on laboratory experiments[55,56]. The initial consolidation pressure $p_0^{ini}$ was chosen 20 times larger than the tensile strength[57] leading to $\beta = 0.05$. Note that in the PST, the slab fails only under tension and thus the absolute value of $p_0$ has no effect on the results, only $\beta p_0$ does. The hardening factor $\xi$ of the slab was chosen based on laboratory experiments of triaxial tests of snow[34,58,59] but could also be derived from strength–density relationships[57,60].

**Table 2 Model parameters**

| Parameter | Slab | | | Weak layer | | |
|---|---|---|---|---|---|---|
| | Exp_01 | Exp_02 | Exp_03 | Exp_01 | Exp_02 | Exp_03 |
| Density $\rho$ (kg m$^{-3}$) | 279 | 255 | 159 | 100 | 100 | 100 |
| Young's modulus $E$ (MPa) | 12 | 8.5 | 2 | 1 | 1 | 1 |
| Poisson's ratio $\nu$ | 0.3 | 0.3 | 0.3 | 0.3 | 0.3 | 0.3 |
| Thickness $D$ (cm) | 70 | 75 | 26 | 7.5 | 15 | 1 |
| Initial consolidation pressure $p_0^{ini}$ (kPa) | 93 | 75 | 24 | 11 | 22 | 4 |
| Tension/compression ratio $\beta$ | 0.05 | 0.05 | 0.05 | 0.2 | 0.2 | 0.2 |
| Friction coefficient $M$ | 0.5 | 0.5 | 0.5 | 0.5 | 0.5 | 0.5 |
| Hardening factor $\xi$ | 30 | 30 | 30 | 0.25 | 0.07 | 0.05 |
| Softening factor $\alpha$ | — | — | — | 15 | 250 | 5 |

For the weak layer, its thin and fragile character prevents mechanical testing to measure relevant mechanical properties. Hence, the shape of our Cohesive Cam Clay yield surface was based on laboratory experiments of weak snow failure[11] and simulations based on X-ray microtomography[31]. The initial consolidation pressure $p_0$ and the softening factor $\alpha$ (which controls the fracture energy) were obtained by matching the critical crack length and propagation speed in the experiments and the simulations. The hardening factor $\xi$ determines the amount of volumetric collapse and was thus evaluated from PST experiments. The tension/compression ratio $\beta$ was chosen equal to 0.2[11]. The density of the weak layer was chosen equal to 100 kg m$^{-3}$ (lower range of measurements reported in Jamieson and Johnston[61]).

For both the slab and the weak layer, the friction coefficient $M$ was chosen equal to 0.5[11,41] and the Poisson's ratio equal to 0.3[57]. Model parameters are given in Table 2.

**Numerical model.** Material deformation is characterized by the strain measure. Assuming there is a deformation map $\phi(\mathbf{X}, t)$ that maps undeformed coordinate $\mathbf{X}$ to a deformed coordinate $\mathbf{x}$, the deformation gradient $\mathbf{F}$ is defined as $\partial\phi/\partial\mathbf{X}$. Our physical model assumes finite strain elastoplasticity, where $\mathbf{F}$ is decomposed into elastic and plastic parts as $\mathbf{F} = \mathbf{F}^E\mathbf{F}^P$. The Hencky strain $\epsilon$ also provides a convenient description of elastic deformation. It is related to $\mathbf{F}^E$ as $\epsilon = \frac{1}{2}\log\left(\mathbf{F}^E(\mathbf{F}^E)^T\right)$. We can write the singular value decomposition of $\mathbf{F}^E$ as $\mathbf{F}^E = \mathbf{U}^E\mathbf{\Sigma}^E\mathbf{V}^E$ following the convention from Irving et al.[62]. It can be shown that $\mathbf{U}^E$ diagonalizes $\epsilon$[63]. Consequently in the principal space, we have

$$\hat{\epsilon}_i = \log\Sigma^E_{ii} \tag{6}$$

where $\hat{\epsilon}_i$ are the eigenvalues of $\epsilon$.

For the constitutive relation, we adopt the St. Venant–Kirchhoff energy density as in Klar et al.[63]:

$$\Psi(\mathbf{F}^E) = \mu\,\mathrm{tr}(\epsilon^2) + \frac{1}{2}\lambda\mathrm{tr}(\epsilon)^2, \tag{7}$$

where $\mu$ and $\lambda$ are Lamé parameters. In terms of the Hencky strain, it implies the following stress strain relationship:

$$\boldsymbol{\tau} = \mathcal{C} : \epsilon, \tag{8}$$

where $\mathcal{C}$ is the fourth-order elastic modulus tensor and $\boldsymbol{\tau}$ is the Kirchhoff stress tensor. If we denote the principal stress as $\hat{\boldsymbol{\tau}}$ and represent $\hat{\boldsymbol{\tau}}$ and $\hat{\epsilon}$ with vectors, $\mathcal{C}$ reduces to a matrix $\mathbf{C}$ and we may write

$$\hat{\boldsymbol{\tau}} = \mathbf{C}\hat{\epsilon}. \tag{9}$$

$\mathbf{C}$ is given by $\mathbf{C} = 2\mu\mathbf{I} + \lambda\mathbf{1}\mathbf{1}^T$, where $\mathbf{I}$ is the identity matrix, $\mathbf{1}$ is the all ones vector. We may further define $\mathbf{D} = \mathbf{C}^{-1}$ so that $\hat{\epsilon} = \mathbf{D}\hat{\boldsymbol{\tau}}$.

An elastoplastic model is not complete without a flow rule. Our model follows the same principle as the MCC model and obeys an associated flow rule. Recall the multiplicative decomposition $\mathbf{F} = \mathbf{F}^E\mathbf{F}^P$, we have the elastic right Cauchy–Green strain tensor $\mathbf{C}^E$ and the elastic left Cauchy–Green strain tensor $\mathbf{b}^E$ as $\mathbf{C}^E = (\mathbf{F}^E)^T\mathbf{F}^E$ and $\mathbf{b}^E = \mathbf{F}^E(\mathbf{F}^E)^T$[64]. Furthermore, $\mathbf{C}^P = (\mathbf{F}^P)^T\mathbf{F}^P$ denotes the plastic right Cauchy–Green strain tensor. The associative plastic flow rule is given by Simo[65] and Simo and Meschke[66]

$$-\frac{1}{2}\mathcal{L}_{\mathbf{v}}\mathbf{b}^E = \dot{\gamma}\frac{\partial y}{\partial\boldsymbol{\tau}}\mathbf{b}^E, \tag{10}$$

$$\dot{\gamma} \geq 0, y \leq 0, \dot{\gamma}y = 0, \tag{11}$$

where $\mathcal{L}_{\mathbf{v}}\mathbf{b}^E = \mathbf{F}\frac{\partial}{\partial t}(\mathbf{C}^P)^{-1}\mathbf{F}^T$ is the Lie derivative of $\mathbf{b}^E$, $\dot{\gamma}$ is the plastic consistency parameter and Eq. (11) are the Kuhn–Tucker conditions. The associativity corresponds to the direction choice of $\frac{\partial y}{\partial\boldsymbol{\tau}}$. This choice is also known as the principle of maximum plastic dissipation[64], leading to a plastic flow that

maximizes the plastic dissipation rate. We refer to the derivation by Klár et al.[63] for more detailed discussion of the associative flow rule and non-associative flow rule. Note that combined with the flow rule, our plastic model perfectly satisfies the second law of thermodynamics, thus energy will never increase during the simulation.

The return mapping is the discrete equivalent of solving for the strain that lies inside the yield surface and satisfies the flow rule. Following the derivations from Simo and Meschke[66] and Klar et al.[63], we can show (see Supplementary Note 1) that if a trial elastic strain $\hat{\epsilon}^{tr} = \log\Sigma^E$ is computed assuming there is no plasticity, the return mapping corresponds to solving for $\hat{\epsilon}^{n+1}$ that satisfies

$$\hat{\epsilon}^{tr} - \hat{\epsilon}^{n+1} = \Delta\gamma\frac{\partial y}{\partial\hat{\boldsymbol{\tau}}} \tag{12}$$

subject to $y\left(\hat{\boldsymbol{\tau}}\left(\hat{\epsilon}^{n+1}\right)\right) \leq 0$, where $\hat{\boldsymbol{\tau}}$ is related to $\hat{\epsilon}^{n+1}$ through the elastic modulus tensor (see Eq. (9)). We note that return mapping for associative plasticity is equivalent to solving the following optimization problem:

$$\hat{\boldsymbol{\tau}} = \arg\,\min_{\hat{\boldsymbol{\tau}}}\left\|\hat{\boldsymbol{\tau}} - \hat{\boldsymbol{\tau}}^{tr}\right\|^2_{\mathbf{C}^{-1}}$$

$$\mathrm{s.t.}\,y(\hat{\boldsymbol{\tau}}) = 0, \tag{13}$$

where $\hat{\boldsymbol{\tau}} = \mathbf{C}\hat{\epsilon}^{n+1}$ and $\hat{\boldsymbol{\tau}}^{tr} = \mathbf{C}\hat{\epsilon}^{tr}$ are the projected stress and trial stress in the principal space respectively, and $\|\hat{\boldsymbol{\tau}}\|^2_{\mathbf{C}^{-1}} = \frac{1}{2}\hat{\boldsymbol{\tau}}^T\mathbf{C}^{-1}\hat{\boldsymbol{\tau}}$. It can be verified that the optimality condition of the Lagrangian reveals Eq. (12).

We now consider the 2D case. If the trial stress lies inside the yield surface, snow deforms elastically and we set $\hat{\epsilon}^{n+1} = \hat{\epsilon}^{tr}$. Otherwise, we need to project the stress onto the yield surface by solving the nonlinear equations of $\hat{\epsilon}^{n+1}$. Note that we have three unknowns $\hat{\epsilon}_1^{n+1}, \hat{\epsilon}_2^{n+1}$ and $\Delta\gamma$ and three equations

$$\mathbf{f} = \hat{\epsilon}^{n+1} + \Delta\gamma\frac{\partial y}{\partial\hat{\boldsymbol{\tau}}}\left(\hat{\epsilon}^{n+1}\right) - \hat{\epsilon}^{tr} = \mathbf{0}, \tag{14}$$

$$y = y\left(\hat{\boldsymbol{\tau}}\left(\hat{\epsilon}^{n+1}\right)\right) = 0. \tag{15}$$

We can efficiently solve this system through a classical Newton's method. We have found that the iterative process usually converges within 2–3 iterations.

In 3D we could follow the same procedure with 2D. However that results in a nonlinear system with four unknowns. Inverting a $4 \times 4$ Hessian is much more expensive than inverting a $3 \times 3$ one. We develop a novel procedure that reduces the number of unknowns to 3. Since $y$ can be written in terms of $p$ and $q$, the goal is to parametrize $\hat{\boldsymbol{\tau}}$ in the same subspace. First, we rewrite Eq. (14) as

$$\mathbf{D}\hat{\boldsymbol{\tau}} + \Delta\gamma\frac{\partial y}{\partial\hat{\boldsymbol{\tau}}}\left(\hat{\epsilon}^{n+1}\right) - \hat{\epsilon}^{tr} = \mathbf{0}. \tag{16}$$

Let's use $\alpha_i$ to denote some unknown coefficients. From Eq. (2), we know $\hat{\boldsymbol{\tau}} = \alpha_1\mathbf{S} + \alpha_2\mathbf{1}$. Furthermore, it can be shown that both $\mathbf{S}$ and $\mathbf{1}$ are eigenvectors of $\mathbf{D}$, therefore $\mathbf{D}\hat{\boldsymbol{\tau}} = \alpha_3\mathbf{S} + \alpha_4\mathbf{1}$. Differentiating the yield function reveals that $\frac{\partial y}{\partial\hat{\boldsymbol{\tau}}} = \alpha_5\mathbf{S} + \alpha_6\mathbf{1}$. As a result, we have $\hat{\epsilon}^{tr} = \alpha_7\mathbf{S} + \alpha_8\mathbf{1}$. Plugging in Eq. (2) gives $\hat{\boldsymbol{\tau}} = \alpha_9\hat{\epsilon}^{tr} + \alpha_{10}\mathbf{1}$. We can further define the deviatoric trial strain $\hat{\epsilon}^{dev} = \hat{\epsilon}^{tr} - \frac{1}{3}\mathrm{tr}(\hat{\epsilon}^{tr})\mathbf{1}$ to get $\hat{\boldsymbol{\tau}} = \alpha_{11}\frac{\hat{\epsilon}^{dev}}{\|\hat{\epsilon}^{dev}\|} + \alpha_{12}\mathbf{1}$. Thus $\hat{\boldsymbol{\tau}}$ must lie in the plane spanned by $\hat{\epsilon}^{dev}$ and $\mathbf{1}$. We can solve for $\alpha_{11}$ and $\alpha_{12}$ using the definition of $p$ and $q$. The result is given by

$$\hat{\boldsymbol{\tau}} = -p\mathbf{1} + q\sqrt{\frac{2}{3}}\frac{\hat{\epsilon}^{dev}}{\|\hat{\epsilon}^{dev}\|}. \tag{17}$$

When we substitute this result into the optimization problem 13 and treat $p$ and $q$

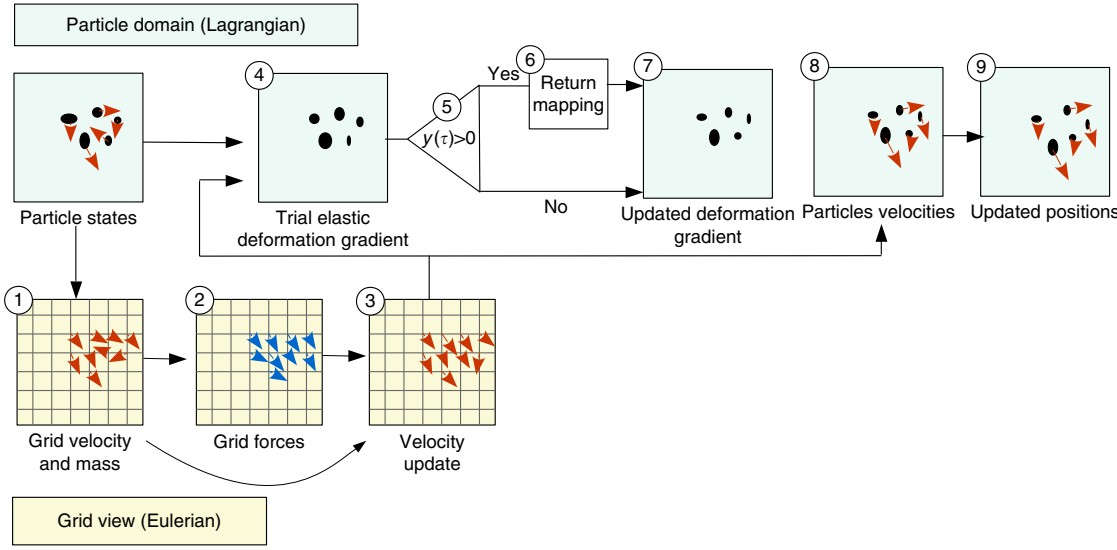

**Fig. 5** Overview of the MPM algorithm

as unknowns, the optimal condition becomes a nonlinear system in terms of $p$, $q$, and $\Delta\gamma$ which can be solved efficiently with a $3 \times 3$ Newton's method. Once $p$ and $q$ are solved, we compute $\hat{\tau}$ using Eq. (17) and $\hat{\epsilon}^{n+1} = \mathbf{D}\hat{\tau}$.

In the following, we show how we track the volumetric plastic strain. The return mapping essentially extract extra deformation from the trial state and push it into the plastic strain. Since our hardening law depends only on $\epsilon_V^P = \log(J^P) = \log(\det(\mathbf{F}^P))$, we only need to store this single variable instead of the full $\mathbf{F}^P$. It is updated as

$$\left[\epsilon_V^P\right]^{n+1} = \left[\epsilon_V^P\right]^n + \left(\mathrm{tr}(\hat{\epsilon}^{\mathrm{tr}}) - \mathrm{tr}(\hat{\epsilon}^{n+1})\right) \tag{18}$$

at the end of the return mapping algorithm. For the modified softening law of the porous weak snow layer, it is updated as

$$[\eta]^{n+1} = [\eta]^n + \alpha \left|\mathrm{tr}(\hat{\epsilon}^{\mathrm{tr}}) - \mathrm{tr}(\hat{\epsilon}^{n+1})\right| \tag{19}$$

where $\alpha$ is the softening factor.

**Material point method**. The material deformation changes according to conservation of mass, momentum, and the elastoplastic constitutive model:

$$\frac{D\rho}{Dt} = 0, \rho \frac{D\mathbf{v}}{Dt} = \nabla \cdot \boldsymbol{\sigma} + \rho\mathbf{g}, \boldsymbol{\sigma} = \frac{1}{J}\frac{\partial\Psi}{\partial\mathbf{F}}\mathbf{F}^E, \tag{20}$$

where $J = \det(\mathbf{F})$ and $\boldsymbol{\sigma} = \boldsymbol{\tau}/J$ is the Cauchy stress tensor. MPM[21] consists in using particles (material points) to track mass, momentum, and deformation gradient. The Lagrangian character of these quantities facilitates the discretization of the mass conservation equation as well as the acceleration term in the momentum conservation equation. However, the lack of mesh connectivity between particles complicates the calculation of spatial derivatives of the stress tensor ($\nabla \cdot \boldsymbol{\sigma}$). Hence, this is done by using a regular background Eulerian grid mesh and interpolation functions over this grid in the standard FEM manner using the weak form. Hence, in MPM, there is no inherent need for Lagrangian mesh connectivity and, in a large deformation framework, MPM implicitly handle fractures and collisions. This is essential to simulate the dynamics of materials which evidence many topological changes such as snow.

We closely follow the explicit MPM algorithm from Stomakhin et al.[24] with a symplectic Euler time integrator. The primary difference is the elastoplastic constitutive model regarding how stress is computed and processed under the plastic flow. We describe here the update procedure in which each step is illustrated in Fig. 5. The first step consists in transferring mass and velocities from particles to the grid using a generalization of the Fluid-Implicit-Particle (FLIP) method[67] for solid mechanics. The mass is transferred using the weighting functions $m_\mathbf{i}^n = \sum_p m_p w_{\mathbf{i}p}^n$ with $\mathbf{i} = (i, j, k)$ is the grid cell index. Velocity is also transferred to the grid, but weighting with $w_{\mathbf{i}p}^n$ does not result in momentum conservation. Instead, we use normalized weights for velocity $\mathbf{v}_\mathbf{i}^n = \sum_p \mathbf{v}_p^n m_p w_{\mathbf{i}p}^n / m_\mathbf{i}^n$. Then, we compute grid forces and update grid velocities (steps 2 and 3). A trial elastic deformation gradient is computed (step 4) and the yield condition is checked (step 5). If $y(\boldsymbol{\tau}) > 0$, we perform return mapping algorithm (step 6) to update deformation gradient (step 7). If $y(\boldsymbol{\tau}) \leq 0$ we keep the trial deformation gradient. Finally,

we compute particle velocities (step 8) and update particle positions (step 9). We refer to Jiang et al.[68] for all other steps in the MPM time stepping algorithm.

**Data availability**. Data supporting the plots of the manuscript and other results of this study are available from the corresponding author upon request.

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

## Acknowledgements

This project was partially funded by the Swiss National Science Foundation (grant number IZK0Z2_174275). J.G. acknowledges financial support from the Swiss National Science Foundation (grant number PZ00P2_161329). C.J. acknowledges financial support from the National Science Foundation (grant number NSF IIS-1755544).

## Author contributions

C.J. developed the MPM code and the general snow constitutive model with T.G. under the guidance of J.T. T.G. and J.G. developed the new strain softening model for the weak layer and performed the simulations. A.H. performed the field measurements, developed,

and applied the PTV algorithm. J.G. obtained the EPFL-UCLA collaboration funding and wrote the paper. All authors were involved in editing the manuscript.

## Additional information

**Competing interests:** The authors declare no competing interests.

