## [Peer Review File · Nature Communications]

Reviewers' comments:

Reviewer #1 (Remarks to the Author):

In "Dynamic anticrack propagation in snow" Gaume et al. develop and apply a new elastoplastic fracture model to snow failure. The new model is applied in 2 and 3 dimensions. Specifically, the model is used to simulate Propagation Saw Test results and slope-scale failures.

Unfortunately, I do not have the expertise to fully critique the numerical model. With regard to snow avalanche fracture, my background is as a practitioner, field scientist, and I have some background with linear elastic fracture mechanics; therefore, I can only assess the motivation, importance, general structure, results, and validation efforts. I strongly suggest that this manuscript also be reviewed by an elastoplastic fracture expert, not necessarily from the snow community.

My overall impression of this manuscript is very positive. Given a thorough vetting by an elastoplastic fracture expert to ensure that the model foundations are sound, I would recommend publication subject to a few minor changes. I also agree with the authors that this model and the results are applicable to other viscoelastic materials, such as concrete and foams, not just snow. Thus, the study should appeal to a wide audience.

The spatial and dynamic nature of the model developed (which I suggest naming something less generic than "Elastoplastic model") is novel. Snow fracture models have been inordinately devoted to crack initiation and propagation, often in a single dimension, e.g. Dave McClung's work over the past four decades. Although, it is important to recognize McClung's contributions to the treatment of snow as a quasi-brittle strain-softening material. Far less attention has been paid to stopping fracture and to modeling features such as en echelon cracks and avalanche crown faces. From a practical perspective, the processes that cause these features are of critical importance. For example, in new snow, fracture initiation and some propagation is common, however large avalanches often do not occur because the slab breaks up, as accurately portrayed by the elastoplastic model presented here. Previous fracture models fail to capture this behavior in new snow, which demonstrates the limitations of such narrowly focused models.

As the authors point out, phenomena such as en echelon cracks have been observed and documented in the literature (van Herwijnen, 2005; Gauthier and Jamieson, 2010), but the models have been lacking in terms of adequate explanation of these observations. The elastoplastic model presented here shows strain softening and other plastic effects, which the authors demonstrate are essential for modeling avalanche formation.

I really appreciate the 2-D remote triggering, zoom on crown fracture, and 3-D remote triggering simulations (Supplementary Videos 3-5). These videos are the highlights of this manuscript and I have not seen a snow avalanche modeled with a particle package like Houdini before.

I have a few general suggestions.

1) Elaborate and speculate on the differences in crown opening behavior between the PSTs and the slope-scale simulation. It's stated that the model reconciles the differences between transverse fractures for these two types of boundary conditions, but no further explanation is given.

2) Discuss the shortcomings of the slope scale simulation in Video 5. I see several. For example, flanks never meet the crown at a 90 deg angle like that. The crown is never completely perpendicular to the slope when viewed from overhead. It is often jagged (Perla, 1971) or convex with the apex at the highest upslope point. Please comment on the why the crown does not have a realistic upslope profile.

3) Give a detailed explanation of how the Houdini package was used to create Video 5. What sort of boundary conditions were used? How do the slab and weak layer properties vary spatially?

4) In Video 5, it looks as though the slope was carefully constructed with a dip in the middle to give a profile with crown taper at both ends. Please comment on this geometry.

5) Explain how the plastic zone from the model (i.e. the red area in the PST videos) compares and differs from the Fracture Process Zone (Bažant et al., 2003; Sigrist, 2006).

I've included a few other suggestions as comments in the PDF.*

Bažant, Z.P., Zi, G. and McClung, D., 2003. Size effect law and fracture mechanics of the triggering of dry snow slab avalanches. *Journal of Geophysical Research*, 108(B2): 2119.

Gauthier, D. and Jamieson, B., 2010. On the sustainability and arrest of weak layer fracture in whumpfs and avalanches. 2010 International Snow Science Workshop: 224-231.

Perla, R.I., 1971. The slab avalanche. Department of Meteorology, Ph.D.

Sigrist, C., 2006. Measurement of fracture mechanical properties of snow and application to dry snow slab avalanche release. (Diss. No. 16736).

van Herwijnen, A., 2005. Fractures in weak snowpack layers in relation to slab avalanche release. Department of Civil Engineering: 315.

*Editorial Note: In their review of the first version of this manuscript, reviewer 1 added some comments to the manuscript file. These comments, excluding minor textual revisions, have been copied into this Peer Review File.

-Line 49: with F^E and P^P as elastic and plastic components, I assume?

-Fig 2 legend: How does this compare to the Fracture Process Zone (e.g. Sigrist 2006)? Sigrist, C. (2006), Measurement of fracture mechanical properties of snow and application to dry snow slab avalanche release, doi: 10.3929/ethz-a-005282374.

-Line 176: What does your new model tell us about the cause of the difference in crown opening between PSTs and slope scale failures? For instance, is the crack opening from the top of the PST is caused by different boundary conditions when compared to full size avalanches? What other differences are you seeing between the PST and slope-scale model runs, e.g. crack velocity, vertical displacement?

-Ref 28: oops

-Methods section title: videos need to label collapse phase as "ACP" instead of "CP" to be consistent w/ text

-Methods section line 7: I think you should cite by name here, i.e. Greene et al. 2016 (note the newer revision)

-Methods section Fig 1: any comment on why the saw direction is teeth forward compared with the original suggestion (Gauthier and Jamieson, 2008) of teeth away cut direction. Gauthier, D., and B. Jamieson (2008), Evaluation of a prototype field test for fracture and failure propagation propensity in weak snowpack layers, *Cold Regions Science and Technology*, 51, 87-97, doi: 10.1016/j.coldregions.2007.04.005.

Reviewer #2 (Remarks to the Author):

General comments:

This paper introduces a new fully 3D elastoplastic formulation of the constitutive behavior of snow which account for the strong softening behavior associated to the collapse of the weak layer

triggering slab avalanches. This model is inspired by Modified Cam Clay models widely used in soil mechanics. The authors successfully transfer it to a new unusual material. Combined with the use of a Material Point Method the authors show the ability of their model to:

- reproduce Propagation Saw Test (field experiments)
- produce very convincing remote avalanche triggering 2D and 3D simulation
- capture both the triggering and the flow of slab avalanches.

In addition, the paper is well written, very clear and well supported by its supplementary material that gives nearly all required information to understand the model details.

I think the paper is suitable for publication after addressing the following few comments.

Specific comments:

1. My main question about this paper lies in the practical choice of the model parameters and the sensitivity of the model to the different parameters. If I well understand the details given in the supplementary material, some of the parameters (E , ν , βp_0 , ...) are chosen from the literature and some are calibrated on the PST experiments (p_0 , α and ξ in particular). This makes the comparison with PST experiments a little less convincing as they are not fully validation tests but also partly calibration tests. Could you comment on that? As far as I know, mechanical properties of snow found in the literature show important discrepancies. Did you perform any sensitivity analysis of your model with respect to its parameters?

2. In the title you advertise for "anticrack" but in the end you use an elastoplastic model in which there is no crack in the sense of displacement discontinuities. Why didn't you use "compaction band" instead of "anticrack"? This is a minor remark but I don't see the need to introduce rather unphysical cracking mechanisms at the continuum mechanics scale here.

Detailed comments :

- I.43: I would suggest "a hybrid Eulerian-Lagrangian method suitable to deal with large strains"?

- I.43: A reference to the MPM code used is missing. At least refer to the supplement where details can be found.

- I.49: "Hooke's law"

- equation (1): You should mention somewhere your sign convention: compressions counted positive and volume decrease for $\epsilon_{_v} < 0$.

- Figure 1: I would suggest you add the initial point of the curves (c) and (d) on Figure (b) which is the same as (1*) if I am right. I would also suggest you add the different states (initial, (0), (1), (1*), (2*)) on in Figure (a). By doing so, I think it will help the reader better understand the model.

- I.55: Shouldn't it be $q = \sqrt{3/2} s$?

- equation (5): How does η modifies the yield surface? Is it through p as suggested in Figure 1 (b)?

- I.85-87: The following sentence is not clear to me: "Hence, when stresses in the weak layer reach the yield surface, the introduction of the norm of $\dot{\epsilon}_V^P$ in Eq. 5 will lead to volume reduction even under compression for which $\dot{\epsilon}_V^P < 0$." Why $\dot{\epsilon}_V^P < 0$ could lead to volume increase?

- I.103: In the end of your section "Elastoplastic model" I would suggest you summarize all the different model parameters together with their physical meaning.
- I.116: "a critical crack length" unless you can define it. Idem on line 142.
- I.135: How do you compute the elastic wave propagation?
- I.134,135,138,147: $m \cdot s^{-1}$. The dot is missing. Check other occurrences of this typo.
- supplement I.55: Isn't some E exponents missing?
- supplement equation (7): I am sorry, I failed to understand how you go from equation (5) to (7). Could you detail the technical computation?
- supplement I.74: If $\hat{\epsilon}_{1}^{n+1}$ and $\hat{\epsilon}_{2}^{n+1}$ are scalars they should not appear in bold.
- supplement I.111: Is index i corresponding to the grid cell number?

Reviewer #3 (Remarks to the Author):

Comments on the paper " Dynamic anticrack propagation in Snow"

The authors have used plasticity combined softening with to model collapse of weak layer leading to avalanche initiation. This an original contribution and also a new approach which along with use of MPM probably speeds up calculations (no estimates for which have been given) and makes possible 3-D simulations at large scale (This again is a guess as authors have not provided any dimensions of 3-D simulation). In past, R.L.Brown has studied collapse of snow using Carol and Holt model of pore collapse. The authors can look at his work and see if it provides any insights. The authors have not related their constitutive model to structure of snow. This seems to be the weakness of the present model. The intention of the authors at this stage may be to show that the method works and at a later stage bring in the microstructure.

Modeling approach

Hardening and softening in Slab:

It has been assumed that hardening and softening within the material depends basically on volumetric plastic strain along with material bulk modulus K and hardening parameter, ξ , as other parameters.

It is understood that as soon as yield condition is reached and hardening/ softening process starts. Will same hardening/ softening law be applicable under shear force? How does failure occur under pure shear?

Hardening and softening in weak layer:

Definition of t_c corresponding to $\epsilon_V^P=0$, and process for the condition when $t \leq t_c$ is not understood.

Line 84 If $\epsilon_V^P=0$ then the material has become incompressible. Although authors have not given any physical meaning to it, this seems to be the condition when snow has become ice. Subsequently, hardening takes place (hardening of incompressible ice).

Material Modeling Method (MPM)

For numerical modeling, authors have used material point method (MPM).

Density of material points in slab and weak snow layers is not discussed. Is there any effect of material point density on solution? What is the optimum material point density?

How material points are treated on failure/ softening is not discussed?

Actual boundary of the domain and location of material points is different. How boundary

conditions are defined?

Experimental methods and model parameters:

Authors have conducted three PST experiments and manual snow profiles. They have recorded various input snow parameters (density, slab thickness, weak layer thickness) terrain parameters (slope angle) and outcome of the experiments (critical crack length, position of slab fracture, PST outcome).

Geometry and density of the slab are directly used for model whereas other parameters are derived indirectly (through expressions or data given in literature) with density as the primary input.

For weak layer only thickness is the measured parameter. No justification of selecting a density of 100 Kg./m³ for weak layer and other parameters has been given.

Softening and hardening parameters are estimated by matching experimental and model results of PST.

Results

In first part of the results experimental observations (vertical slab displacements, crack propagation in weak layer, full or partial crack propagation, weak layer collapse) on three PSTs are discussed.

In second part of the results observations in simulations (anticrack propagation on flat terrain, collapse wave speed etc.) and their comparison with experimental results (bending phase, anticrack propagation, frictional sliding) are discussed.

Critical crack length values are not compared.

It appears that some of the model parameters are extracted through comparison of simulated and experimental results of the three PSTs and in results again same simulation results are compared with same experiments. Hence, it will be nice to see if the simulation results are compared with some new experimental data.

Discussion and conclusions

In discussion section reproduction of crack branching in simulations has been claimed however the same has not been presented in the results section.

In the final video, although phenomenon of avalanche release is shown, how correct are the dimensions of slab released. At the top tensile failure is responsible for slab. What causes the failure of the slab on the sides?

Does the collapse/failure of weak layer, cause any shear failure at interface?

An excellent effort by the authors to simulate avalanche initiation and flow using a single framework, however, there is a need to develop a methodology to estimate model parameters based on snow properties/ characteristics for effective application of the model.

Response to reviewer's comments

In the following, we provide (in blue) detailed point-by-point answers to the comments raised by the reviewers (in black, *italic*). In addition, changes made to the manuscript are highlighted in red. Our main changes concern:

- a new larger 3D slope simulation with more realistic flanks in the release zone (comment of reviewer #1).
- a new subsection in the Results to describe the slope simulation results including a figure showing the release zone and deposit of the avalanche as well as a more detailed description of the setup of slope scale simulations in the supplement (comments of reviewers #1 and #3).
- the presentation of the mathematical derivation of our return mapping algorithm from the plastic flow rule (first derived by Simo and Meschke, 1993) in the form of a supplement (comment of reviewer #2)
- a deeper discussion on the choice of model parameters (comments of reviewers #2 and #3).
- a new supplementary note to describe how failure occurs in pure shear in the weak layer (comment of reviewer #3)
- a new supplementary note to evidence that our model is consistent with mesh refinement (comment of reviewer #3).

Response to Referee #1

In Dynamic anticrack propagation in snow Gaume et al. develop and apply a new elastoplastic fracture model to snow failure. The new model is applied in 2 and 3 dimensions. Specifically, the model is used to simulate Propagation Saw Test results and slope-scale failures.

Unfortunately, I do not have the expertise to fully critique the numerical model. With regard to snow avalanche fracture, my background is as a practitioner, field scientist, and I have some background with linear elastic fracture mechanics; therefore, I can only assess the motivation, importance, general structure, results, and validation efforts. I strongly suggest that this manuscript also be reviewed by an elastoplastic fracture expert, not necessarily from the snow community.

My overall impression of this manuscript is very positive. Given a thorough vetting by an elastoplastic fracture expert to ensure that the model foundations are sound, I would recommend publication subject to a few minor changes. I also agree with the authors that this model and the results are applicable to other viscoelastic materials, such as concrete and foams, not just snow. Thus, the study should appeal to a wide audience.

The spatial and dynamic nature of the model developed (which I suggest naming something less generic than Elastoplastic model) is novel. Snow fracture models have been inordinately devoted to crack initiation and propagation, often in a single dimension, e.g. Dave McClungs work over the past four decades. Although, it is important to recognize McClungs contributions to the treatment of snow as a quasi-brittle strain-softening material. Far less attention has been paid to stopping fracture and to modeling features such as en echelon cracks and avalanche crown faces. From a practical perspective, the processes that cause these features are of critical importance. For example, in new snow, fracture initiation and some propagation is common, however large avalanches often do not occur because the slab breaks up, as accurately portrayed by the elastoplastic model presented here. Previous fracture models fail to capture this behavior in new snow, which demonstrates the limitations of such narrowly focused models.

As the authors point out, phenomena such as en echelon cracks have been observed and documented in the literature (van Herwijnen, 2005; Gauthier and Jamieson, 2010), but the models have been lacking in terms of adequate explanation of these observations. The elastoplastic model presented here shows strain softening and other plastic effects, which the authors demonstrate are essential for modeling avalanche formation.

I really appreciate the 2-D remote triggering, zoom on crown fracture, and 3-D remote triggering simulations (Supplementary Videos 3-5). These videos are the highlights of this manuscript and I have not seen a snow avalanche modeled with a particle package like Houdini before.

We thank Referee #1 for his positive comments on our manuscript and for very constructive suggestions that helped us to improve the quality of our paper (see below). Given the very positive response to our slope-scale simulations by the reviewer (also by reviewer #2) we now provide a new figure showing the 3D avalanche simulation (release and deposit) in a new subsection describing slope-scale simulations. Concerning the section “Elastoplastic model”, we renamed it as “**Large strain elastoplastic model**” to be less generic, as suggested.

Figure 1: 3D slope-scale simulation (supplementary video 7) just after avalanche release showing an arc crown line as well as jagged flanks and staunchwall (a) and after the flow of the avalanche (b).

I have a few general suggestions.

Comment 1). *Elaborate and speculate on the differences in crown opening behavior between the PSTs and the slope-scale simulation. Its stated that the model reconciles the differences between transverse fractures for these two types of boundary conditions, but no further explanation is given.*

Answer to comment 1). We agree that this important aspect could be discussed in more detail. As clearly seen in video 5 (crown fracture in slow motion), it appears that the crown fracture is a secondary process that occurs after the crack in the weak layer has passed the crown. Hence, after crack propagation and collapse of the weak layer, the weak layer has a purely frictional shear behavior leading to a pure tension stress state (no bending) in the slab which has started to slide on the weak layer where the slope angle is large enough. The tensile stress is thus maximum at the interface with the weak layer where the fracture initiates. In contrast, in the PST, the fracture in the slab arrests the propagation of the crack in the weak layer. As a consequence, the bending of the slab due to the collapse of the weak layer on the left side of the crack tip (Fig. 2c) induces larger stresses at the top of the slab where the fracture initiates in that case. We added the following paragraph to our discussion section to elaborate on this point as suggested by the reviewer:

“(…) Hence, our model reconciles contradictory observations of slab fractures from small scale field tests (top to bottom) and from real avalanches (bottom to top). We suggest that the main reason for this discrepancy is related to the slope angle gradient at the crown and the frictional sliding of the slab. In the supplementary video 5 (crown fracture in a 2D slope simulation), it appears that the crown fracture is a secondary process occurring after the crack in the weak layer has passed. Hence, the tensile stress induced by slab bending was not sufficient to induce a tensile fracture, very likely due to the large propagation speed which reduces bending as suggested by Gaume et al. (2015). However, after crack propagation and collapse, the weak layer has a frictional shear behavior leading to a pure tension stress state (no bending) in the slab (Gaume et al., 2015) which has started to slide on the weak layer only where the slope is steep enough. This induces very large tensile stresses in the slab which are maximum at the interface with the weak layer where the fracture initiates. In contrast, in PST experiments and simulations (Fig. 2c), the bending of the slab induced by the crack in the weak layer created with the saw is sufficient to lead to slab fracture and the arrest of crack propagation in the weak layer. In that case, stresses are larger at the top of the slab where the fracture initiates (Gaume et al., 2015).”

Comment 2). *Discuss the shortcomings of the slope scale simulation in Video 5. I see several. For example, flanks never meet the crown at a 90 deg angle like that. The crown is never completely perpendicular to the slope when viewed from overhead. It is often jagged (Perla, 1971) or convex with the apex at the highest upslope point. Please comment on the why the crown does not have a realistic upslope profile.*

Answer to comment 2). In our simulations, the flanks meet the crown at a 90 degrees angle because the failure reached the boundary condition of our system on the sides. This is due to the system dimensions and the homogeneity of the system properties. This is now stated in the simulation description in the supplementary note 1. We agree with the reviewer that this aspect is not observed in the field. As we believe that we can reach even more realistic release characteristics, we performed a new 3D simulation of a larger slope including spatial variability of slab thickness (video 7 with 30 million particles / 5 days of simulation on 24 i7 Intel CPU cores). We added a figure in the paper showing the release zone and deposit of this avalanche. As one can see, the release zone is now extremely realistic and not influenced by the boundary conditions of our system.

Concerning the perpendicular character of our crown fracture, we agree with the reviewer that some convex crown fractures have been observed but we are not sure if this can be generalized. As seen in the following pictures (Fig. 2), the crown fracture looks very close to be perpendicular to the slope. It has also been reported in numerous publications that the crown is perpendicular to the bed surface (McClung and Schweizer (2006), page 80 of McClung and Schaefer (2006) and Perla (1971)). We suppose that cases with non-perpendicular fracture shapes are the result of a strong layering in the slab. In our simulations the slab is uniform.

[Redacted]

Finally, concerning the “jagged” character, we believe Perla (1971) refers to the flanks (“Right and left flank line - Side section of the fracture line usually jagged”), not to the crown as shown in figure 1.4 of Perla (1971). Interestingly, our new 3D avalanche simulation also shows jagged flanks and staunchwall (video 7 and figure 1 left).

To account for the reviewer’s comments, we introduced a new subsection (see below) in the Result section to describe our slope scale simulations:

Slope-scale simulations Two and three dimensional slope-scale simulations of remote avalanche triggering were performed (setup description and videos in supplementary note 1). In both slope simulations (2D and 3D), the crack propagation speed was around 60 m.s^{-1} and the crown fracture was almost perpendicular to the bed surface as reported by Perla (1971) and McClung and Schweizer (2006). Furthermore, the slab fracture at the crown of the avalanche (upslope section of the fracture line) started branching from the bottom of the slab at the interface with the weak layer (supplementary video 5), in contrast to the PST simulation and experiment n° 3 in which it started branching from the top. In 3D, the simulated release zone (figure 3, supplementary video 7)

has commonly observed characteristics (Perla, 1971): an arc crown line as well as jagged flanks (side sections of the fracture line) and staunchwall (bottom section of the fracture line). Crown failure occurs in tension while flank and staunchwall failures occur in shear. Finally, the cross-slope propagation was approximately twice slower than up-slope propagation.

Comment 3). *Give a detailed explanation of how the Houdini package was used to create Video 5. What sort of boundary conditions were used? How do the slab and weak layer properties vary spatially?*

Answer to comment 3). We added in the description of the supplementary videos, a more detailed explanation of the boundary conditions and characteristics of our slope simulations. We only fixed the bottom of the weak layer in PST simulations. The rest is free. For 2D slope simulations, the bottom of the weak layer, the left and right extremities of the slab are fixed. In 3D simulations, the sides are also fixed. See also figures 4 and 5 (reply to reviewer 3).

Comment 4). *In Video 5, it looks as though the slope was carefully constructed with a dip in the middle to give a profile with crown taper at both ends. Please comment on this geometry.*

Answer to comment 4). This geometry was chosen to mimic a concave slope with a maximum snow depth in the middle of the path. It was reported by Vontobel et al. (2013) that this type of slope shape was most commonly associated with avalanches. This is now commented in the description of our slope-scale simulations in the supplement.

Comment 5). *Explain how the plastic zone from the model (i.e. the red area in the PST videos) compares and differs from the Fracture Process Zone (Baant et al., 2003; Sigrist, 2006).*

Answer to comment 5). The red area in PST and slope-scale simulations represent particles where plasticity occurred in the weak layer ($\epsilon_V^P \neq 0$) and thus cannot directly compare to the fracture process zone (FPZ). However, our model has similarities with the fracture mechanical model of Bazant (cohesive zone model, Bazant et al., 2003). This similarity can now be seen in more details as we have now added a new supplement with a shear test simulation. In pure shear (figure 6 below or supplementary figure 3), the shear stress vs shear strain curve is very similar to that assumed by Bazant et al. (2003). Specifically, the so-called critical sliding displacement has the same order of magnitude ($\delta_s = 3.5$ mm in Bazant et al. (2003), $\delta_s \sim 2$ mm in McClung (1977); Gaume et al. (2013) and $\delta_s \sim 2$ mm in our simulation). Note, however that in the model of Bazant et al. (2003), the weak layer is considered as perfectly rigid, whereas it is elastoplastic in our model which prevents a direct comparison. This leads to the introduction of a characteristic length of the system associated with the elastic mismatch between the slab and the weak layer ($\Lambda = \sqrt{E' D D_{wl} / G_{wl}}$, Gaume et al., 2013, 2017). This length characterizes the decrease of stresses close to the crack tip and is typically found between 0.3 and 1.5 m (Gaume et al., 2017) which is larger than the size of FPZs measured by Sigrist (2006). To account for the reviewer's comment, we added the following paragraph in the new supplementary note 3:

We describe and discuss the mechanical behavior of the weak layer in shear by simulating an unconfined shear-test (simulation setup corresponds to experiment n° 2 with a wall horizontal velocity of 0.01 m.s^{-1} , supplementary figure 3). After reaching the shear strength of the weak layer, significant softening is observed followed by weak layer collapse (almost 20% of weak layer thickness). Then, a pure frictional behavior with a residual shear strength is observed. This behavior is very similar to what has been reported in laboratory experiments of shear failure of snow (McClung, 1977) and what has been assumed in interfacial shear models (Bazant et al., 2003; Gaume et al., 2013). In particular, the so-called critical sliding displacement (displacement δ_s associated to the softening zone) has the same order of magnitude as in Bazant et al. (2003) and McClung (1977) ($\delta_s = 3.5$ mm in Bazant et al. (2003), $\delta_s \sim 2$ mm in McClung (1977); Gaume et al. (2013) and $\delta_s \sim 2$ mm in our simulation).

Detailed comments

line 9 *allows simulating* → *simulates*. Thanks. This was corrected as suggested

line 40 *with F^E and F^P as elastic and plastic components, I assume?* Yes. This is now precised in the text.

figure 2 *How does this compare to the Fracture Process Zone (e.g. Sigrist 2006)? Sigrist, C. (2006), Measurement of fracture mechanical properties of snow and application to dry snow slab avalanche release Please see answer to comment 5.*

line 150 *allowed to reproduce → reproduced. Thanks. This was corrected.*

line 176 *What does your new model tell us about the cause of the difference in crown opening between PSTs and slope scale failures? For instance, is the crack opening from the top of the PST is caused by different boundary conditions when compared to full size avalanches? What other differences are you seeing between the PST and slope-scale model runs, e.g. crack velocity, vertical displacement? Thanks for this remark. Concerning differences between slope-scale and PST simulations, the main difference concerns slab fracture on which we now elaborate according to the reviewer's suggestion (comment 1). Average crack propagation speeds in slope-scale simulations were found to be larger than in PST simulation (~ 60 m/s) but not incompatible with what has been measured in the field with the PST (Gaume et al., 2015) and in real avalanches (Van Herwijnen and Schweizer, 2011; Hamre et al., 2014). As Gaume et al. (2015) showed that crack propagation speed was increasing significantly with increasing slope angle, we suspect that this difference is mostly due to the larger slope angle in slope-scale simulation ($\psi_{max} = 45^\circ$). Finally, we observed that cross-slope propagation was slower than up-slope propagation. These observations are now described in the new paragraph describing slope simulations and discussed in the discussion section (see answers to comment 1 and 2).*

videos *videos need to label collapse phase as "ACP" instead of "CP" to be consistent w/ text Thanks for this remark. Videos were all modified as suggested.*

methods line 7 *I think you should cite by name here, i.e. Greene et al. 2016 (note the newer revision) Thanks. Corrected.*

methods figure 1 *any comment on why the saw direction is teeth forward compared with the original suggestion (Gauthier and Jamieson, 2008) of teeth away cut direction. Gauthier, D., and B. Jamieson (2008), Evaluation of a prototype field test for fracture and failure propagation propensity in weak snowpack layers, Cold Regions Science and Technology, 51, 87-97 Thanks for noticing. We modified the direction of the saw in this drawing to be consistent with the PST guidelines (Gauthier and Jamieson, 2008; Greene et al., 2016).*

Response to Referee #2

General comments

This paper introduces a new fully 3D elastoplastic formulation of the constitutive behavior of snow which account for the strong softening behavior associated to the collapse of the weak layer triggering slab avalanches. This model is inspired by Modified Cam Clay models widely used in soil mechanics. The authors successfully transfer it to a new unusual material. Combined with the use of a Material Point Method the authors show the ability of their model to:

- *reproduce Propagation Saw Test (field experiments)*
- *produce very convincing remote avalanche triggering 2D and 3D simulation*
- *capture both the triggering and the flow of slab avalanches.*

In addition, the paper is well written, very clear and well supported by its supplementary material that gives nearly all required information to understand the model details.

I think the paper is suitable for publication after addressing the following few comments.

We thank Referee #2 for his positive response to our manuscript and for very constructive suggestions that helped us to improve the quality of our manuscript (see below).

Specific comments

Comment 1). *My main question about this paper lies in the practical choice of the model parameters and the sensitivity of the model to the different parameters. If I well understand the details given in the supplementary material, some of the parameters (E , ν , βp_0 , ...) are chosen from the literature and some are calibrated on the PST experiments (p_0 , α and ξ in particular). This makes the comparison with PST experiments a little less convincing as they are not fully validation tests but also partly calibration tests. Could you comment on that? As far as I now, mechanical properties of snow found in the literature show important discrepancies. Did you perform any sensitivity analysis of your model with respect to its parameters?*

Answer to comment 1). Thanks for this remark. We agree with the reviewer that some of the parameters (of the weak layer) were chosen to match PST outputs (critical crack length, propagation speed, collapse height). This is mostly due to the fact that relevant weak layer parameters do not exist in the literature and cannot be directly measured in the field as it would require triaxial testing which is very difficult for very thin and fragile layers. Our main objective here is to show that our model is capable of reproducing all important features observed in field experiments and in slab avalanches which is extremely challenging. However, as mentioned in the concluding remark of our paper, future work is required to make a systematic evaluation of our model parameters using a larger dataset, similar to what was done in van Herwijnen et al. (2016) for the elastic modulus of the slab or from numerical modeling based on X-ray computed tomography (Hagenmuller et al., 2015). To account for the reviewer's comment, this last sentence of our paper was modified to put the stress on this important aspect:

In the future, the parameters of our model should be systematically derived from in-situ measurements and related to snow type and density. The main difficulty lies in the thin and fragile nature of weak layers which prevents efficient mechanical testing such as triaxial tests to evaluate relevant model parameters. Hence, a calibration based on PST results using a larger dataset similar to what was done in van Herwijnen et al. (2016) or an evaluation based on X-ray computed tomography (Hagenmuller et al., 2015) will be required. This would allow to develop a predictive model to mitigate and forecast real scale gravitational hazards by using digital elevation models of real slopes obtained from laser scanning or photogrammetry as input.

Finally, we performed a sensitivity analysis when we calibrated weak layer parameters (p_0 , α and ξ), but this analysis was specific to our PST examples and not broadly relevant and thus out of the scope of the present paper. Note however that part of this sensitivity analysis (effect of the mesh size dx on the change in volumetric plastic strain for different ξ and p_0 values) is now shown in supplementary note 4 to answer a comment of reviewer #3 about the influence of mesh resolution.

Comment 2). *In the title you advertise for "anticrack" but in the end you use an elastoplastic model in which there is no crack in the sense of displacement discontinuities. Why didn't you used "compaction band" instead of "anticrack"? This is a minor remark but I don't see the need to introduce rather unphysical cracking mechanisms at the continuum mechanics scale here*

Answer to comment 2). The coupling of our elastoplastic model with the Material Point Point Method actually leads to displacement discontinuities in zones where softening occurs. This is one of the major advantage of MPM. These displacement discontinuities can be clearly observed in slope-scale simulations. It occurs in mode I and II in the slab (tensile fracture at the crown, shear fracture at the flanks) but also in as a mixed-mode anticrack in the weak layer (mode -I/II, compression-shear). We agree that this fracture process is peculiar and is rarely observed in classical materials but the concept of anticrack is well defined and has been the focus of several studies, from e.g. snow science (Heierli et al., 2008, 2011) to porous rocks mechanics (Fletcher and Pollard, 1981), submarine landslides (Locat et al., 2014) and deep earthquakes (Green et al., 1990; Green, 2007). In our paper, the concept of anticrack is defined in the two first sentences: "Cohesive porous materials under compression often evidence volumetric collapse leading to localization of compaction or compacting shear bands. This peculiar process is generally referred to as anticrack (...)". Hence, given the arguments above and the definition at the beginning of our paper, we prefer to keep our title.

Detailed comments

1.43 I would suggest "a hybrid Eulerian-Lagrangian method suitable to deal with large strains"? Agreed. This was modified as suggested.

1.43 A reference to the MPM code used is missing. At least refer to the supplement where details can be found. Thanks for this remark. We referred to Sulsky et al. (1995) as well as in the methods.

1.49 "Hooke's law" Thanks. This was corrected

equation (1) You should mention somewhere your sign convention: compressions counted positive and volume decrease for $\epsilon_V < 0$. Thanks. The sign convention was added as suggested.

Figure 1 I would suggest you add the initial point of the curves (c) and (d) on Figure (b) which is the same as (1*) if I am right. I would also suggest you add the different states (initial, (0), (1), (1*), (2*)) on in Figure (a). By doing so, I think it will help the reader better understand the model. Thanks for this remark. We changed the numbering in Figs. 1b, 1c and 1d to account for the reviewer's comment. However, we tried adding these points on Figure 1a but it makes it too busy and more complicated since two additional ellipses need to be drawn (points (2) and (3*)).

1.55 Shouldn't it be $q = \sqrt{3/2 \mathbf{s} : \mathbf{s}}$? Yes absolutely. Thanks for noticing. This was changed as follows and we give the detail of this expression in 2D and 3D:

(...) given by $q = (3/2 \mathbf{s} : \mathbf{s})^{1/2}$ (so that $q = |\tau_1 - \tau_2|$ for 2D and $q = \sqrt{\frac{1}{2} \left((\tau_1 - \tau_2)^2 + (\tau_2 - \tau_3)^2 + (\tau_3 - \tau_1)^2 \right)}$ for 3D.)

equation (5) How does η modifies the yield surface? Is it through p as suggested in Figure 1 (b)? Yes η changes the yield surface through a decrease of p_0 . Due to the introduction of η , the yield surface shrinks until it reaches the origin of the $q - p$ curve (corresponding to $p_0 = 0$). This was clarified in the text by modifying the following sentence, which also account for the next comment of the reviewer:

Hence, when stresses in the weak layer reach the yield surface, the introduction of the norm of ϵ_V^P in Eq. 5 will lead to **softening (through a decrease of p_0)** even under compression for which $\epsilon_V^P < 0$. The yield surface thus shrinks until it corresponds to a point at the origin of the $p - q$ space.

1.85-87 The following sentence is not clear to me: "Hence, when stresses in the weak layer reach the yield surface, the introduction of the norm of ϵ_V^P in Eq. 5 will lead to volume reduction even under compression for which $\epsilon_V^P < 0$." Why $\epsilon_V^P < 0$ could lead to volume increase? Thanks for this remark. We agree. In fact, using the new weak layer model, $\epsilon_V^P < 0$ leads to softening, even under compression, not (yet) to volume reduction. The collapse (volume reduction) occurs after stress reduction in the weak layer ($p_0 = 0$) when using the classical hardening law without cohesion because of the compressive stress due to the slab. This is now clarified in the text as follows:

Hence, when stresses in the weak layer reach the yield surface, the introduction of the norm of ϵ_V^P in Eq. 5 will lead to **softening (through a decrease of p_0)** even under compression for which $\epsilon_V^P < 0$. The yield surface thus shrinks until it corresponds to a point at the origin of the $p - q$ space. In addition, cohesion is removed by setting $\beta = 0$ when $\epsilon_V^P = 0$ which ensures continuity. After reaching this point, the yield surface is free to expand according to the classical hardening law (Eq. 4), **leading to volume reduction (collapse) due the weight of the slab (blue arrows in Fig. 2b)** and then to a purely frictional/compaction behavior.

1.103 In the end of your section "Elastoplastic model" I would suggest you summarize all the different model parameters together with their physical meaning. Thanks for this suggestion. We added the following sentence to our manuscript to account for this remark:

Let us summarize here the different model parameters and their physical meaning: p_0 is the consolidation pressure and represents the compressive strength of the material, β is the ratio between tensile and compressive strength and characterizes cohesion, M is the slope of the Critical State Line (CSL) and characterizes the friction of the material, K is the bulk elastic modulus, ξ is the hardening coefficient and characterizes the brittleness of the material (a large ξ makes snow more brittle) and α is the softening factor which controls the fracture energy of the weak layer.

1.116 a critical crack length" unless you can define it. Idem on line 142. Thanks. Corrected as suggested.

1.135 How do you compute the elastic wave propagation? We made a rough estimation using the E-wave velocity $c^E = \sqrt{E/\rho}$. This is now mentioned in the manuscript as follows:

This speed is significantly lower than the speed of elastic waves $c^E = \sqrt{E/\rho}$ (...)

1.134,135,138,147 m.s-l. The dot is missing. Check other occurrences of this typo. Thanks. This was corrected.

supplement 1.55 supplement 1.55: Isn't some E exponents missing? We checked but could not find a missing exponent here. The E part is included in $\mathbf{F} = \mathbf{F}^E \mathbf{F}^P$. Please find below how the Lie derivative is obtained:

$$\begin{aligned} \mathbf{b}^E &= \mathbf{F}^E (\mathbf{F}^E)^T = \mathbf{F} (\mathbf{F}^P)^{-1} (\mathbf{F} (\mathbf{F}^P)^{-1})^T \\ &= \mathbf{F} ((\mathbf{F}^P)^{-1} (\mathbf{F}^P)^{-T}) \mathbf{F}^T \\ &= \mathbf{F} ((\mathbf{F}^P)^T (\mathbf{F}^P))^{-1} \mathbf{F}^T \\ &= \mathbf{F} (\mathbf{C}^P)^{-1} \mathbf{F}^T, \end{aligned}$$

where we defined $\mathbf{C}^P = (\mathbf{F}^P)^T (\mathbf{F}^P)$. Then we take the time derivative:

$$\dot{\mathbf{b}}^E = \frac{\partial \mathbf{F}}{\partial t} (\mathbf{C}^P)^{-1} \mathbf{F}^T + \mathbf{F} \frac{\partial ((\mathbf{C}^P)^{-1})}{\partial t} \mathbf{F}^T + \mathbf{F} (\mathbf{C}^P)^{-1} \frac{\partial (\mathbf{F}^T)}{\partial t}.$$

The second term is the Lie derivative $\mathcal{L}_{\mathbf{v}} \mathbf{b}^E = \mathbf{F} \frac{\partial}{\partial t} (\mathbf{C}^P)^{-1} \mathbf{F}^T$

supplement equation (7) I am sorry, I failed to understand how you go from equation (5) to (7). Could you detail the technical computation? This derivation is indeed not trivial. It was published initially in Simo and Meschke (1993) and then adapted by Klár et al. (2016) for Drucker-Prager elastoplasticity for sand simulations. We only gave references here but given the comment of the reviewer, we now also present this complex derivation with our notations. This is done in the form of a new supplementary note which is also presented below:

The return mapping algorithm is the discrete equivalent to solving for a strain that satisfies the plastic flow rule in Equation 10 of the Methods section and that lies in the CamClay yield surface. In this section first we outline the method of Simo and Meschke (1993) to derive the discrete equations from their continuous versions. This procedure starts by assuming there is no plastic flow and a return mapping algorithm is derived from the flow equations that shows how to project back to the yield surface if the assumption of no plastic flow is invalid.

Consider the evolution of \mathbf{b}^E from time t^n to time $t^{n+1} = t^n + \Delta t$. We consider this evolution per particle, and thus it is useful to take a Lagrangian view. We outline the notation used in the Lagrangian view in the Methods section of the paper. Specifically useful here is the flow map $\phi : \Omega^0 \times [0, T] \rightarrow \mathbb{R}^d$, and its relation to the deformation gradient $\mathbf{F} = \frac{\partial \phi}{\partial \mathbf{X}}$. Define the time t^n configuration of the material as $\Omega^{t^n} = \{\tilde{\mathbf{x}} = \phi(\mathbf{X}, t^n) \text{ for some } \mathbf{X} \in \Omega^0\}$ and define $\tilde{\phi} : \Omega^{t^n} \times [t^n, T] \rightarrow \mathbb{R}^d$ as $\tilde{\phi}(\tilde{\mathbf{x}}, t) = \phi(\phi^{-1}(\tilde{\mathbf{x}}, t^n), t)$. Intuitively, $\tilde{\phi}$ defines the deformation as if the time t^n configuration Ω^{t^n} of the material is the reference

configuration, rather than Ω^0 as in the standard Lagrangian view. This is some times called an updated Lagrangian view. While the deformation gradient \mathbf{F} defines the deformation from the initial configuration (Ω^0) to the time t configuration (Ω^t), the Jacobian $\tilde{\mathbf{F}} = \frac{\partial \tilde{\phi}}{\partial \tilde{\mathbf{x}}}$ defines the deformation from the time t^n configuration (Ω^{t^n}) to the time t configuration (Ω^t), where $t \geq t^n$. Also these are related as $\mathbf{F} = \tilde{\mathbf{F}}\mathbf{F}^n$, or more precisely $\mathbf{F}(\mathbf{X}, t) = \tilde{\mathbf{F}}(\phi(\mathbf{X}, t^n), t)\mathbf{F}(\mathbf{X}, t^n)$ for all $\mathbf{X} \in \Omega^0$.

Define $\mathbf{b}^{E*} = \tilde{\mathbf{F}}^{-1}\mathbf{b}^E\tilde{\mathbf{F}}^{-T}$. Let us consider the difference between the evolution of \mathbf{b}^{E*} and \mathbf{b}^E in absence of plasticity at time $t^n < t < t^{n+1}$. By the definition of \mathbf{b}^{E*} , $\frac{D\mathbf{b}^{E*}}{Dt} = -2\dot{\gamma}\tilde{\mathbf{F}}^{-1}\mathbf{G}\tilde{\mathbf{F}}\mathbf{b}^{E*}$, with $\mathbf{G} = \frac{\partial y}{\partial \boldsymbol{\tau}}$, therefore in absence of plasticity \mathbf{b}^{E*} is constant since $\frac{D\mathbf{b}^{E*}}{Dt} = \mathbf{0}$. In contrast, $\mathbf{b}^E|_t = \tilde{\mathbf{F}}|_t \mathbf{b}^E|_{t^n} \tilde{\mathbf{F}}^T|_t$ in the same case. In other words, \mathbf{b}^{E*} is constant except for the effect of plasticity, but at the same time \mathbf{b}^E would also be stretched by the flow. This isolation of the plastic part allows for a more intuitive discretization. Specifically, if we let $\mathbf{H} = (\dot{\gamma}\tilde{\mathbf{F}}^{-1}\mathbf{G}\tilde{\mathbf{F}})|_{t^{n+1}}$, we have that \mathbf{b}^{E*} approximately satisfies the ODE: $\frac{D\mathbf{Y}}{Dt} = -2\mathbf{H}\mathbf{Y}$, with $\mathbf{Y}|_{t^n} = \mathbf{b}^E|_{t^n}$, and we can approximate $\mathbf{b}^{E*}|_{t^{n+1}}$ by $\mathbf{Y}|_{t^{n+1}} = \exp(-2\Delta t\mathbf{H})\mathbf{b}^E|_{t^n} = \exp(-2\Delta\gamma\tilde{\mathbf{F}}^{-1}\mathbf{G}\tilde{\mathbf{F}})|_{t^{n+1}} \mathbf{b}^E|_{t^n}$ where $\Delta\gamma = \Delta t \dot{\gamma} \geq 0$ will be used to enforce the constraint $y(\boldsymbol{\tau}(\mathbf{b}^E|_{t^{n+1}})) \leq 0$. Multiplying the approximation by $\tilde{\mathbf{F}}|_{t^{n+1}}$ on the left and $\tilde{\mathbf{F}}^T|_{t^{n+1}}$ on the right, and recalling the definition of \mathbf{b}^{E*} , we obtain

$$\begin{aligned} \mathbf{b}^E|_{t^{n+1}} &= \tilde{\mathbf{F}}|_{t^{n+1}} \mathbf{b}^{E*}|_{t^{n+1}} \tilde{\mathbf{F}}^T|_{t^{n+1}} \\ &\approx \tilde{\mathbf{F}}|_{t^{n+1}} \exp(-2\Delta\gamma\tilde{\mathbf{F}}^{-1}\mathbf{G}\tilde{\mathbf{F}})|_{t^{n+1}} \mathbf{b}^E|_{t^n} \tilde{\mathbf{F}}^T|_{t^{n+1}} \\ &= \tilde{\mathbf{F}}|_{t^{n+1}} \tilde{\mathbf{F}}^{-1}|_{t^{n+1}} \exp(-2\Delta\gamma\mathbf{G})|_{t^{n+1}} \tilde{\mathbf{F}}|_{t^{n+1}} \mathbf{b}^E|_{t^n} \tilde{\mathbf{F}}^T|_{t^{n+1}} \\ &= \exp(-2\Delta\gamma\mathbf{G})|_{t^{n+1}} \tilde{\mathbf{F}}|_{t^{n+1}} \mathbf{b}^E|_{t^n} \tilde{\mathbf{F}}^T|_{t^{n+1}}. \end{aligned}$$

Using the notation $\tilde{\mathbf{b}}^E = \tilde{\mathbf{F}}|_{t^{n+1}} \mathbf{b}^E|_{t^n} \tilde{\mathbf{F}}^T|_{t^{n+1}}$, we are looking for a solution pair $\Delta\gamma$ and $\mathbf{b}^E|_{t^{n+1}}$ such that

$$\mathbf{b}^E|_{t^{n+1}} = \exp(-2\Delta\gamma\mathbf{G}(\boldsymbol{\tau}(\mathbf{b}^E|_{t^{n+1}}))) \tilde{\mathbf{b}}^E, \quad (1)$$

and constraint $y(\boldsymbol{\tau}(\mathbf{b}^E|_{t^{n+1}})) \leq 0$ is satisfied. Note that $\tilde{\mathbf{b}}^E$ is the elastic strain we would get without the effect of plasticity. For example if $y(\boldsymbol{\tau}(\tilde{\mathbf{b}}^E)) \leq 0$, then $\Delta\gamma = 0$ and $\mathbf{b}^E|_{t^{n+1}} = \tilde{\mathbf{b}}^E$ is the trivial solution pair and there is no plastic flow. In this sense, we can see that $\tilde{\mathbf{b}}^E$ can be considered as the trial elastic state obtained without any plastic flow. If this does not satisfy the constraint, $\Delta\gamma$ and $\mathbf{b}^E|_{t^{n+1}}$ must be defined to ‘project’ $\tilde{\mathbf{b}}^E$ to $\mathbf{b}^E|_{t^{n+1}}$.

We use this process to define the projection. $\tilde{\mathbf{F}}\mathbf{F}^E$ is considered the trial elastic state, one obtained in the absence of plastic flow. Thus, $\tilde{\mathbf{b}}^E = \tilde{\mathbf{F}}\mathbf{b}^E\tilde{\mathbf{F}}^T$ and we seek the solution of Equation 1 to define the projection to $\mathbf{b}^E|_{t^{n+1}}$. This can be done most easily by considering the singular value decomposition of $\tilde{\mathbf{F}}\mathbf{F}^E$.

If the singular value decomposition of $\tilde{\mathbf{F}}\mathbf{F}^E$ is given by $\tilde{\mathbf{F}}\mathbf{F}^E = \mathbf{U}\tilde{\boldsymbol{\Sigma}}\mathbf{V}^T$, then $\tilde{\mathbf{b}}^E = \mathbf{U}\tilde{\boldsymbol{\Sigma}}^2\mathbf{U}^T$. It can be shown that \mathbf{U} diagonalizes $\mathbf{G}(\boldsymbol{\tau}(\mathbf{b}^E|_{t^{n+1}}))$ and $\mathbf{b}^E|_{t^{n+1}}$ (i.e. $\mathbf{G}(\boldsymbol{\tau}(\mathbf{b}^E|_{t^{n+1}})) = \mathbf{U}\hat{\mathbf{G}}(\boldsymbol{\Sigma})\mathbf{U}^T$, and $\mathbf{b}^E|_{t^{n+1}} = \mathbf{U}\boldsymbol{\Sigma}^2\mathbf{U}^T$ with $\hat{\mathbf{G}} = \frac{\partial y}{\partial \boldsymbol{\tau}}$ and $\hat{\boldsymbol{\tau}} = \mathbf{U}^T\boldsymbol{\tau}(\mathbf{b}^E|_{t^{n+1}})\mathbf{U}$), then we may write (1) as

$$\begin{aligned} \mathbf{U}\boldsymbol{\Sigma}^2\mathbf{U}^T &= \exp(-2\Delta\gamma\mathbf{U}\hat{\mathbf{G}}(\boldsymbol{\Sigma})\mathbf{U}^T) \mathbf{U}\tilde{\boldsymbol{\Sigma}}^2\mathbf{U}^T \\ &= \mathbf{U} \exp(-2\Delta\gamma\hat{\mathbf{G}}(\boldsymbol{\Sigma})) \tilde{\boldsymbol{\Sigma}}^2\mathbf{U}^T. \end{aligned}$$

Multiplying both sides of the previous equation by \mathbf{U}^T on the left and by \mathbf{U} on the right, and taking log results in

$$2 \log(\boldsymbol{\Sigma}) = -2\Delta\gamma\hat{\mathbf{G}}(\boldsymbol{\Sigma}) + 2 \log(\tilde{\boldsymbol{\Sigma}}). \quad (2)$$

The model that we choose uses the Hencky-strain as a measure of deformation. By defining

$$\hat{\epsilon}^{\text{tr}} := \log \tilde{\Sigma} \quad \text{and} \quad \hat{\epsilon}^{n+1} := \log \Sigma, \quad (3)$$

we may simplify and rearrange Equation (2)

$$\hat{\epsilon}^{\text{tr}} - \hat{\epsilon}^{n+1} = \Delta\gamma \hat{\mathbf{G}}. \quad (4)$$

This is our discrete flow rule. In the return mapping algorithm, we want to solve for $\hat{\epsilon}^{n+1}$ satisfies Equation (4) subject to the constraint

$$y(\hat{\boldsymbol{\tau}}(\hat{\epsilon}^{n+1})) \leq 0. \quad (5)$$

Solving Equation (4) and (5) can be seen as a ray-ellipse intersection problem due to the ellipsoid shape of our CamClay yield surface.

supplement 1.74 *If $\hat{\epsilon}_1^{n+1}$ and $\hat{\epsilon}_2^{n+1}$ are scalars they should not appear in bold. Thanks for noticing it. It has been corrected.*

supplement 1.111 *Is index i corresponding to the grid cell number? Thanks for this remark. In fact i should be bold as it is a vector representing the grid cell index $\mathbf{i} = (i, j, k)$. It was corrected and \mathbf{i} is now described.*

Response to Referee #3

We thank Referee #3 for very useful comments that helped us to improve the quality of our manuscript (see below).

Comment 1). *Comments on the paper Dynamic anticrack propagation in Snow The authors have used plasticity combined softening with to model collapse of weak layer leading to avalanche initiation. This an original contribution and also a new approach which along with use of MPM probably speeds up calculations (no estimates for which have been given) and makes possible 3-D simulations at large scale (This again is a guess as authors have not provided any dimensions of 3-D simulation).*

Answer to comment 1). Thank you for these remarks. We now added in the supplement the time required for PST and slope-scale simulations which were performed on a workstation which has 24 i7 Intel CPU cores and 128 Gb of RAM. 2D PST simulations need approximately 30 minutes (~ 80000 particles, 120 fps), 2D slope simulations need approximately 4 hours (~ 2 million particles, 48 fps), 3D slope simulations need between 2 and 5 days of simulation depending the system size ($\sim 10 - 30$ million particles, 48 fps). System dimensions and boundary conditions are now described in detailed in the supplementary note 1 (see also figures 4 and 5 below).

Comment 2). *In past, R.L.Brown has studied collapse of snow using Carol and Holt model of pore collapse. The authors can look at his work and see if it provides any insights.*

Answer to comment 2). We thank the reviewer for this suggestion as we were not aware of this work. In the study of Brown, a model of pore collapse is used to study snow compaction and hardening whereas we study here the collapse and strain-softening behavior of persistent weak layers. However, this interesting study of Brown is relevant to evaluate the hardening factor of our slab and was thus referred to in our Methods section.

Comment 3). *The authors have not related their constitutive model to structure of snow. This seems to be the weakness of the present model. The intention of the authors at this stage may be to show that the method works and at a later stage bring in the microstructure.*

Answer to comment 3). We agree with the reviewer that we did not explicitly account for snow microstructure in our continuum model. However, we have a general snow constitutive model with hardening in compression and softening in tension/shear to model the snow slab and a different post peak behavior with softening in compression to model anticrack propagation in the weak layer. We thus implicitly account for different snow microstructures as snow slabs generally consist of small rounded grains or precipitation particles, while weak layers typically consist of more angular grains, such as depth hoar or surface hoar. Indeed, at this point we mostly wanted to show that our model is capable of reproducing typical behavior observed and measured in snow.

Comment 4). *Modeling approach Hardening and softening in Slab: It has been assumed that hardening and softening within the material depends basically on volumetric plastic strain along with material bulk modulus K and hardening parameter, as other parameters. It is understood that as soon as yield condition is reached and hardening/ softening process starts. Will same hardening/ softening law be applicable under shear force? How does failure occur under pure shear?*

Answer to comment 4). As we have a fully mixed-mode yield surface (q is the Von Mises equivalent stress which characterizes the amount of shear), our hardening law applies for any loading direction. Since we have an associative plastic flow rule (Equation 10 of the Methods), we project according to the normal of the yield surface ($\partial y/\partial \tau$). Thus, the hardening or softening character depends if the trial elastic stress state is on the right of left of the apex of the ellipse in Fig. 1, respectively. In other words, hardening occurs if $p > (1 - \beta)p_0/2$ leading to volume reduction and softening if $p < (1 - \beta)p_0/2$ leading to volume increase. As a consequence, in pure shear ($p = 0$), the material is softening.

Comment 5). *Hardening and softening in weak layer: Definition of t_c corresponding to $VP=0$, and process for the condition when t_c is not understood. Line 84 If $VP=0$ then the material has become incompressible. Although authors have not given any physical meaning to it, this seems to be the condition when snow has become ice. Subsequently, hardening takes place (hardening of incompressible ice).*

Answer to comment 5). We did not state that the material becomes incompressible if $\epsilon_V^P = 0$ which would be incorrect. It does not mean that snow has become ice either. Physically, it means that the bonds in the weak layer are broken and that grain rearrangement will take place leading to the collapse of the weak layer due to the compressive weight of the slab. After the collapse, snow has no cohesion and has a purely frictional behavior as observed in field experiments (van Herwijnen and Heierli, 2009). We believe that the confusion comes from the statement of the similarity to the frictional behavior of dry sand. Hence, this similarity was removed and we improved the physical interpretation of our softening law as follows:

(...) leading to purely frictional/compaction behavior (blue arrows in Fig. 1b). **Physically, our softening rule reproduces bond breaking in the weak layer and subsequent grain rearrangement leading to volumetric collapse due to the compressive weight of the slab (van Herwijnen and Jamieson, 2005).**

Concerning t_c , we now clarified that t_c is the time corresponding to complete softening, i.e. $\epsilon_V^P = 0$ and $p_0 = 0$ (state (2*) in Fig. 1d).

Comment 6). *Material Modeling Method (MPM) For numerical modeling, authors have used material point method (MPM). Density of material points in slab and weak snow layers is not discussed. Is there any effect of material point density on solution? What is the optimum material point density? How material points are treated on failure/ softening is not discussed?*

Answer to comment 6). For PST simulations, we used a point density of 30'000 points per m^2 ($dx = 0.01$ m, 3 particles per cell). We found that this density did not influence the solution, in particular because the fracture energy of the weak layer is regularized (softening factor α proportional to dx). We present below (figure 3) a curve showing that our solution (here the change in volumetric plastic strain $1 - J_P = 1 - \det(\mathbf{F}^P)$ which characterizes the amount of collapse) in a pure shear simulation of the configuration of experiment 2 does not depend on mesh resolution.

This curve was added in a new supplement to assess the consistency of our model with respect to the mesh resolution. In addition, the particle density was added in the simulation description.

Figure 3: Influence of mesh resolution dx , hardening factor ξ and consolidation pressure p_0 on $1 - J_P$.

Concerning how material points are treated on failure/softening: failure and softening (or hardening) induces a change in deformation gradient and stress as a result of the softening (or hardening) law and return mapping algorithm. This is described in the Methods section (Tracking volumetric plastic strain and Material Point Method).

Comment 7). *Actual boundary of the domain and location of material points is different. How boundary conditions are defined?*

Answer to comment 7). In all simulations, the initial position of MPM particles were sampled using Poisson Disk distribution. Concerning boundary conditions, for PST simulations, the bottom of the weak layer is fixed. The rest is free. For 2D slope simulations, the bottom of the weak layer, the left and right extremities of the slab are fixed. In 3D simulations, the sides are also fixed. We added a better description of material points distribution and boundary conditions in the new supplement on simulation setup description. In particular, two figures (figure 4 and 5 below) were made to describe the dimensions and boundary conditions of our simulated system. We used only a ‘fixed’ type of boundary condition for which the velocity is set to zero ($\mathbf{v} = \mathbf{0}$).

Figure 4: System dimensions for the 2D slope simulation. Triangles represent fixed boundary conditions.

Figure 5: System dimensions for the 3D slope simulations (videos 6 and 7). For supplementary video 6, $L = 18$ m, $H = 7$ m and $W = 10$ m. For supplementary video 7, $L = 25$ m, $H = 10$ m and $W = 15$ m. The bottom of the weak layer and the greyed zones (left, right, top and bottom side walls) are fixed boundary conditions.

Comment 8). *Experimental methods and model parameters: Authors have conducted three PST experiments and manual snow profiles. They have recorded various input snow parameters (density, slab thickness, weak layer thickness) terrain parameters (slope angle) and outcome of the experiments (critical crack length, position of slab fracture, PST outcome). Geometry and density of the slab are directly used for model whereas other parameters are derived indirectly (through expressions or data given in literature) with density as the primary input. For weak layer only thickness is the measured parameter. No justification of selecting a density of 100 Kg./m³ for weak layer and other parameters has been given. Softening and hardening parameters are estimated by matching experimental and model results of PST.*

Answer to comment 8). The density of the weak layer has virtually no influence on the results since the load induced by the slab on the weak layer is significantly larger than the self load of the weak layer in any reported case. Hence, we chose a density of 100 kg/m³ which is in the lower range of measurements for weak snow layers, as reported by Jamieson and Johnston (2001). This is now stated in the Methods section.

Concerning other weak layer parameters, as stated in the text, they were chosen to match PST outputs (critical crack length, propagation speed, collapse height). This is mostly due to the fact that relevant weak layer parameters cannot be directly measured in the field due to its very fragile and thin nature. Our main objective here is to prove that our model is capable of reproducing all important features of a slab avalanche which is extremely challenging. However, as already mentioned in the concluding remark of our paper, future work is required to make a systematic evaluation of our model parameters using a larger dataset, similar to what was done in van Herwijnen et al. (2016) for the elastic modulus of the slab. To account for the reviewer's comment (and for a similar comment of reviewer #2), this last sentence of our paper was modified to put the stress on this important aspect:

In the future, the parameters of our model should be systematically derived from in-situ measurements and related to snow type and density. The main difficulty lies in the thin and fragile nature of weak layers which prevents efficient mechanical testing such as triaxial tests to evaluate relevant model parameters. Hence, a calibration based on PST results using a larger dataset similar to what was done in van Herwijnen et al. (2016) or an evaluation based on X-ray computed tomography (Hagenmuller et al., 2015) will be required. This would allow to develop a predictive model to mitigate and forecast real scale gravitational hazards by using digital elevation models of real slopes obtained from laser scanning or photogrammetry as input.

Comment 9). *Results In first part of the results experimental observations (vertical slab displacements, crack propagation in weak layer, full or partial crack propagation, weak layer collapse) on three PSTs are discussed.*

In second part of the results observations in simulations (anticrack propagation on flat terrain, collapse wave speed etc.) and their comparison with experimental results (bending phase, anticrack propagation, frictional sliding) are discussed. Critical crack length values are not compared.

Answer to comment 9). Thanks for this remark. Critical crack length values are now compared.

Comment 10). *It appears that some of the model parameters are extracted through comparison of simulated and experimental results of the three PSTs and in results again same simulation results are compared with same experiments. Hence, it will be nice to see if the simulation results are compared with some new experimental data.*

Answer to comment 10). As mentioned in comment 8, since the weak layer strength and hardening/softening parameters are calibrated to match PST outcomes (critical crack length, propagation speed, collapse height), we are able to reproduce any field experiment which is why we decided to simulate three PSTs with the three different characteristics observed in the field (full propagation on the flat, full propagation with slab sliding, partial propagation with slab fracture, Gauthier and Jamieson, 2008) to show the promise of our model. Hence, adding more experiments with similar characteristics would not add much to the paper. The next step to have a predictive model would be to systematically evaluate the mechanical parameters of our model from a very large database of field measurements which is now stated more clearly (see answer to comment 8).

Comment 11). *Discussion and conclusions In discussion section reproduction of crack branching in simulations has been claimed however the same has not been presented in the results section.*

Answer to comment 11). We agree. This comment was also made by reviewer #1 and we now have the following new subsection in the Results to describe slope simulations.

Slope-scale simulations Two and three dimensional slope-scale simulations of remote avalanche triggering (set-up description and videos in the supplement) were performed. In both slope simulations (2D and 3D), the crack propagation speed was around 60 m.s^{-1} and the crown fracture was almost perpendicular to the bed surface as reported by Perla (1971) and McClung and Schweizer (2006). Furthermore, the slab fracture at the crown of the avalanche (upslope section of the fracture line) started branching from the bottom of the slab at the interface with the weak layer (supplementary video 5), in contrast to the PST simulation and experiment n° 3 in which it started branching from the top. In 3D, the simulated release zone (figure 3, supplementary video 7) has commonly observed characteristics (Perla, 1971): an arc crown line as well as jagged flanks (side sections of the fracture line) and staunchwall (bottom section of the fracture line). Crown failure occurs in tension while flank and staunchwall failures occur in shear. Finally, the cross-slope propagation was approximately twice slower than up-slope propagation.

Comment 12). *In the final video, although phenomenon of avalanche release is shown, how correct are the dimensions of slab released. At the top tensile failure is responsible for slab. What causes the failure of the slab on the sides?*

Answer to comment 12). Thanks for this remark. All system dimensions are now provided in the supplement (see also figures 4 and 5). Note that the 3D geometry was chosen to mimic a concave slope with a maximum snow depth in the middle of the path. It was reported by Vontobel et al. (2013) that this type of slope shape was most commonly associated with avalanches. On the sides and at the staunchwall, the slab fails in shear. This is now stated in the manuscript in the new section describing slope simulations.

Comment 13). *Does the collapse/failure of weak layer, cause any shear failure at interface?*

Answer to comment 13). Once the weak layer has collapsed, it has indeed a pure shear frictional behavior similar to slip interfacial condition. To account for this comment and also comment 5 highlighting that the physical meaning of our post-peak behavior was perhaps not clearly described, we now provide a supplementary note (presented below) showing the behavior of the weak layer in a pure shear simulation, unambiguously showing the frictional behavior after failure and collapse.

Figure 6: Top: Shear test simulation of the weak layer under slab normal load. The setup corresponds the PST simulation of experiment n° 2. The wall on the left is displaced along the x -direction with a constant speed of 0.01 m.s^{-1} . Middle: Mises equivalent stress q vs shear strain γ in a pure shear test (displacement controlled). Bottom: Axial strain ϵ_a vs shear strain γ .

We describe and discuss the mechanical behavior of the weak layer in shear by simulating an unconfined shear-test (simulation setup corresponds to experiment n° 2 with a wall horizontal velocity of 0.01 m.s^{-1} , supplementary figure 3). After reaching the shear strength of the weak layer, significant softening is observed followed by weak layer collapse (almost 20% of weak layer thickness). Then, the weak layer has a pure frictional behavior with a residual shear strength. This mechanical behavior is very similar to what has been reported in laboratory experiments of shear failure of snow (McClung, 1977) and what has been assumed in interfacial shear models (Bazant et al., 2003; Gaume et al., 2013). In particular, the so-called critical sliding displacement (displacement δ_s associated to the softening zone) has the same order of magnitude as in Bazant et al. (2003) and McClung (1977); Gaume et al. (2013) ($\delta_s = 3.5 \text{ mm}$ in Bazant et al. (2003), $\delta_s = 2 \text{ mm}$ in McClung (1977); Gaume et al. (2013) and $\delta_s \sim 2 \text{ mm}$ in our simulation).

Comment 14). *An excellent effort by the authors to simulate avalanche initiation and flow using a single framework, however, there is a need to develop a methodology to estimate model parameters based on snow properties/ characteristics for effective application of the model.*

Answer to comment 14). Thanks a lot for this positive appreciation of our new model. We also agree that some future work is required so that our model can be used and applied operationally for avalanche mitigation. However, as stated above, we believe that we overcame the main scientific obstacle in this direction, by developing the first unified model able to simulate dynamic anticrack propagation and solid/fluid transition in snow slab avalanches, in 3D and at the slope scale. The next steps concern mostly parameter estimation and calibra-

tion which will be at the heart of future efforts in our group in the next years, but represent a less important contribution than this major scientific achievement.

References

- Bazant, Z., Z. G. and D. McClung, 2003: Size effect law and fracture mechanics of the triggering of dry snow slab avalanches. *J. Geophys. Res.*
- Fletcher, R. C. and D. D. Pollard, 1981: Anticrack model for pressure solution surfaces. *Geology*, **9**(9), 419–424.
- Gaume, J., G. Chambon, N. Eckert, and M. Naaim, 2013: Influence of weak-layer heterogeneity on snow slab avalanche release: Application to the evaluation of avalanche release depths. *J. Glaciol.*, **59**(215), 423–437.
- Gaume, J., G. Chambon, N. Eckert, M. Naaim, and J. Schweizer, 2015: Influence of weak layer heterogeneity and slab properties on slab tensile failure propensity and avalanche release area. *The Cryosphere*, **9**(2), 795–804.
- Gaume, J., A. van Herwijnen, G. Chambon, K. Birkeland, and J. Schweizer, 2015: Modeling of crack propagation in weak snowpack layers using the discrete element method. *The Cryosphere*, **9**, 1915–1932.
- Gaume, J., A. van Herwijnen, G. Chambon, N. Wever, and J. Schweizer, 2017: Snow fracture in relation to slab avalanche release: critical state for the onset of crack propagation. *The Cryosphere*, **11**, 217–228.
- Gauthier, D. and B. Jamieson, 2008: Evaluation of a prototype field test for fracture and failure propagation propensity in weak snowpack layers. *Cold Reg. Sci. Technol.*, **51**(2), 87–97.
- Green, H. W., 2007: Shearing instabilities accompanying high-pressure phase transformations and the mechanics of deep earthquakes. *Proceedings of the National Academy of Sciences*, **104**(22), 9133–9138.
- Green, H. W., T. E. Young, D. Walker, and C. H. Scholz, 1990: Anticrack-associated faulting at very high pressure in natural olivine. *Nature*, **348**(6303), 720–722.
- Greene, E., K. Birkeland, K. Elder, I. McCammon, M. Staples, and D. Sharaf, 2016: Snow, weather, and avalanches: Observational guidelines for avalanche programs in the united states. *American Avalanche Association, Pagosa Springs, Colorado*, (3rd ed).
- Hagenmuller, P., G. Chambon, and M. Naaim, 2015: Microstructure-based modeling of snow mechanics: a discrete element approach. *The Cryosphere*, **9**(5)(1969-1982).
- Hamre, D., R. Simenhois, and K. Birkeland, 2014: Fracture speed of triggered avalanches. In *P. Haegeli (Editor), Proceedings ISSW 2014. International Snow Science Workshop, Banff, Alberta, Canada, 29 September - 3 October 2014*, 174-178.
- Heierli, J., K. Birkeland, R. Simenhois, and P. Gumbsch, 2011: Anticrack model for skier triggering of slab avalanches. *Cold Regions Science and Technology*, **65**(3), 372–381.
- Heierli, J., P. Gumbsch, and M. Zaiser, 2008: Anticrack nucleation as triggering mechanism for snow slab avalanches. *Science*, **321**, 240–243.
- Jamieson, J. and C. Johnston, 2001: Evaluation of the shear frame test for weak snowpack layers. *Ann. Glaciol.*, **32**, 59–69.
- Klár, G., T. Gast, A. Pradhana, C. Fu, C. Schroeder, C. Jiang, and J. Teran, 2016: Drucker-prager elastoplasticity for sand animation. *ACM Trans Graph*, **35**(4), 103:1–103:12.
- Locat, J., S. Leroueil, A. Locat, and H. Lee, 2014: Weak layers: their definition and classification from a geotechnical perspective. In *Submarine mass movements and their consequences*. Springer, 3–12.

- McClung, D., 1977: Direct simple shear tests on snow and their relation to slab avalanche formation. *J. Glaciol.*, **19(81)**, 101–109.
- McClung, D. and P. Schaerer, 2006: *The Avalanche Handbook*. The Mountaineers. Seattle USA.
- McClung, D. and J. Schweizer, 2006: Fracture toughness of dry snow slab avalanches from field measurements. *J. Geophys. Res.*, **111**, F04008.
- Perla, R. I., 1971: *The slab avalanche*. Number 100. University of Utah.
- Sigrist, C., 2006: *Measurements of fracture mechanical properties of snow and application to dry snow slab avalanche release*. PhD thesis, ETH Zürich.
- Simo, J. and G. Meschke, 1993: A new class of algorithms for classical plasticity extended to finite strains. application to geomaterials. *Comput Mech*, **11(4)**, 253–278.
- Sulsky, D., S.-J. Zhou, and H. L. Schreyer, 1995: Application of a particle-in-cell method to solid mechanics. *Computer physics communications*, **87(1-2)**, 236–252.
- van Herwijnen, A., J. Gaume, E. Bair, B. Reuter, K. Birkeland, and J. Schweizer, 2016: Field method for measuring the effective elastic modulus and fracture energy of snowpack layers. *J. Glaciol.*, **In Press**.
- van Herwijnen, A. and J. Heierli, 2009: Measurements of crack-face friction in collapsed weak snow layers. *Geophys. Res. Lett.*, **36**. L23502.
- van Herwijnen, A. and B. Jamieson, 2005: High speed photography of fractures in weak snowpack layers. *Cold Reg. Sci. Technol.*, **43(1-2)**, 71–82.
- Van Herwijnen, A. and J. Schweizer, 2011: Seismic sensor array for monitoring an avalanche start zone: design, deployment and preliminary results. *Journal of Glaciology*, **57(202)**, 267–276.
- Vontobel, I., S. Harvey, and R. S. Purves, 2013: Terrain analysis of skier-triggered avalanche starting zones. *Proceedings of the International Snow Science Workshop ISSW, Grenoble, France, 7 October - 11 October 2013*.

REVIEWERS' COMMENTS:

Reviewer #1 (Remarks to the Author):

The authors have addressed all of my suggestions and I'd recommend publication as is. There are a few grammatical errors that were introduced during the revision, but they are minor and I assume they will be taken care of during copy editing. This is a great contribution and I hope to see it published soon.

NB

Reviewer #2 (Remarks to the Author):

In this resubmission the authors have carefully addressed all the points raised by the referees. I think the paper can be accepted and published as-is.

Though I may still have two following minor comments:

- While justifying the term "anticrack", the authors claims that displacement discontinuities can be observed in MPM simulations. I do not use MPM myself but the displacement field is imposed by interpolation, isn't it? In this case, I was more expecting strain discontinuities (coming from the softening) rather than displacement discontinuities (excepted if the interpolation functions allow displacement discontinuities). Do interpolation functions allow for discontinuities?

- In the softening law, point (2*) corresponds to a yield surface restricted to the point $(p,q)=(0,0)$, however in Figure 1(c) and 1(d), point (2*) corresponds to a non zero p-q stress state. Is this coming from a loss of homogeneity of the snow specimen?

Reviewer #3 (Remarks to the Author):

The reviewer is satisfied with the response of the authors. However, the reviewer would like to know:

Comment 1:

What is the advantage of MPM over other methods of solution.

Comment 4:

It will be appreciated if author can provide a graphical presentation of failure envelope explicitly for their slab and weak layer failure model.

Response to reviewer's comments

Manuscript NCOMMS-17-31723A
Dynamic anticrack propagation in snow
by Gaume, Gast, Teran, van Herwijnen, Jiang

We want to thank again the three reviewers for their additional comments. In the following, we provide (in blue) detailed point-by-point answers to the comments raised by the reviewers (in black, *italic*). In addition, changes made to the manuscript are highlighted in red.

Response to Referee #1

The authors have addressed all of my suggestions and I recommend publication as is. There are a few grammatical errors that were introduced during the revision, but they are minor and I assume they will be taken care of during copy editing. This is a great contribution and I hope to see it published soon.

NB

We thank Referee #1 for the review of our revised manuscript and we are pleased by his positive evaluation of our work. We went through the paper again to correct remaining grammatical errors.

Response to Referee #2

In this resubmission the authors have carefully addressed all the points raised by the referees. I think the paper can be accepted and published as-is.

Though I may still have two following minor comments: - While justifying the term "anticrack", the authors claims that displacement discontinuities can be observed in MPM simulations. I do not use MPM myself but the displacement field is imposed by interpolation, isn't it? In this case, I was more expecting strain discontinuities (coming from the softening) rather than displacement discontinuities (excepted if the interpolation functions allow displacement discontinuities). Do interpolation functions allow for discontinuities?

- In the softening law, point (2) corresponds to a yield surface restricted to the point $(p,q)=(0,0)$, however in Figure 1(c) and 1(d), point (2*) corresponds to a non zero p - q stress state. Is this coming from a loss of homogeneity of the snow specimen?*

We thank Referee #2 for the review of our revised manuscript and for his positive evaluation.

Concerning the first question, the reviewer is right. Our anticrack model is the result of strain softening in the weak layer. "Displacement discontinuities" in our work should rather be referred to as "material separation" and naturally come from the MPM since the grid loses support and allows large deformations (topology change) at plastically softened regions. In this work, interpolation functions do not allow for sharp displacement discontinuities which usually require modifications to the kernel. Note that strong and sharp displacement discontinuities were recently introduced in MPM by the last author of this study in (Hu et al., 2018) by modifying particle-grid transfer using a moving least squares approach.

Concerning the second question, this is correct. In this graph, the stress was averaged in the whole weak layer during an unconfined compression test. However, the failure occurs in a localized manner and particles might not reach the zero p - q stress state at the exact same moment. This loss of homogeneity leads to an averaged stress which is close but not exactly zero after softening. This information was added in the figure caption as follows:

In c and d, p and q in the weak layer (blue curves) do not perfectly reach zero after softening due to a loss of homogeneity (failure localization).

Response to Referee #3

The reviewer is satisfied with the response of the authors. However, the reviewer would like to know:
 Comment 1: What is the advantage of MPM over other methods of solution.

Comment 4: It will be appreciated if author can provide a graphical presentation of failure envelope explicitly for their slab and weak layer failure model.

We thank Referee #3 for the review of our revised manuscript. We are pleased that our replies and changes to the manuscript satisfied the reviewer.

Concerning the first comment, the main advantage of MPM comes from its hybrid Lagrangian-Eulerian character. The Lagrangian character of the particles facilitates the discretization of temporal derivatives while the use of a regular background Eulerian grid facilitates the calculation of the derivatives required for stress-based force evaluation ($\nabla \cdot \sigma$). Hence, in MPM, there is no inherent need for Lagrangian mesh connectivity and, in a large deformation framework, MPM implicitly handle fractures and collisions. This is essential to simulate the dynamics of materials which evidence many topological changes such as snow.

In more detail, compared to FEM, MPM does not need periodical remeshing and remapping state variables and is thus better to deal with large deformations. In addition, since the Lagrangian particles store the information, no remeshing algorithm is required. Compared to pure particle methods, in MPM the computation of the gradients is trivial because of the fixed grid and MPM is computationally less expensive (e.g. in DEM the computational time depends on the number of contacts). Note however that, although MPM has several advantages compared to other numerical methods, it is more expensive in terms of storage because MPM uses mesh as well as particle data.

The following sentence was added in the Methods section in the MPM paragraph (before the algorithm description) to make this point even clearer: **Hence, in MPM, there is no inherent need for Lagrangian mesh connectivity and, in a large deformation framework, MPM implicitly handle fractures and collisions. This is essential to simulate the dynamics of materials which evidence many topological changes such as snow.**

Concerning the second question, as we give the full equation of our failure envelope as follows

$$y(p, q) = q^2(1 + 2\beta) + M^2(p + \beta p_0)(p - p_0), \quad (1)$$

and the parameters p_0 , β and M for the slab and the weak layer for the three experiments, the reader can very easily plot the failure envelopes for these cases. In addition, since the failure envelope is generically plotted on figure 1a of the paper (same shape of failure envelope for the slab and the weak layer), we believe it would not add much to the manuscript to include these specific (and thus not broadly relevant) graphical representations in the manuscript. However, we provide these failure envelopes for the three experiments in this peer-review file (see figure below) which will be made available upon publication as electronic supplementary material:

Figure 1: Graphical representation of the yield surface of the slab and the weak layer for the three experiments.

Just note that, as mentioned in the paper, once the initial failure envelope of the weak layer is reached, the cohesion is set to zero by setting $\beta = 0$ but M remains the same and p_0 evolves according to the soft-

ening/hardening law. For the slab, after reaching the yield surface, β and M remain the same and p_0 evolves according to the hardening rule.

References

Hu, Y., Y. Fang, Z. Ge, Z. Qu, Y. Zhu, A. Pradhana, and C. Jiang, 2018: A moving least squares material point method with displacement discontinuity and two-way rigid body coupling. *ACM Trans. Graph.*